# Development of a methodology to make individual estimates of the precision of liquid chromatography-tandem mass spectrometry drug assay results for use in population pharmacokinetic modeling and the optimization of dosage regimens

Gellért Balázs Karvaly[1]*, Michael N. Neely[2], Krisztián Kovács[1], István Vincze[1], Barna Vásárhelyi[1], Roger W. Jelliffe[2]

1 Department of Laboratory Medicine, Semmelweis University, Budapest, Hungary, 2 Laboratory of Applied Pharmacokinetics and Bioinformatics, Children's Hospital of Los Angeles, Keck School of Medicine, University of Southern California, Los Angeles, California, United States of America

* karvaly.gellert_balazs@med.semmelweis-univ.hu

## Abstract

### Background

The clinical value of therapeutic drug monitoring can be increased most significantly by integrating assay results into clinical pharmacokinetic models for optimal dosing. The correct weighting in the modeling process is 1/variance, therefore, knowledge of the standard deviations (SD) of each measured concentration is important. Because bioanalytical methods are heteroscedastic, the concentration-SD relationship must be modeled using assay error equations (AEE). We describe a methodology of establishing AEE's for liquid chromatography-tandem mass spectrometry (LC-MS/MS) drug assays using carbamazepine, fluconazole, lamotrigine and levetiracetam as model analytes.

### Methods

Following method validation, three independent experiments were conducted to develop AEE's using various least squares linear or nonlinear, and median-based linear regression techniques. SD's were determined from zero concentration to the high end of the assayed range. In each experiment, precision profiles of 6 ("small" sample sets) or 20 ("large" sample sets) out of 24 independent, spiked specimens were evaluated. Combinatorial calculations were performed to attain the most suitable regression approach. The final AEE's were developed by combining the SD's of the assay results, established in 24 specimens/spiking level and using all spiking levels, into a single precision profile. The effects of gross hyperbilirubinemia, hemolysis and lipemia as laboratory interferences were investigated.

**Data Availability Statement:** All relevant data are within the paper and its Supporting Information files.

**Funding:** GBK, KK, IV and BV receive funding from the National Office of Research, Development and Innovation of Hungary (National Bionics Program, Government Decree 1336/2017). The funder had no role in study design, data collection and analysis, decision to publish, or preparation of the manuscript.

**Competing interests:** The authors have declared that no competing interests exist.

## Results

Precision profiles were best characterized by linear regression when 20 spiking levels, each having 24 specimens and obtained by performing 3 independent experiments, were combined. Theil's regression with the Siegel estimator was the most consistent and robust in providing acceptable agreement between measured and predicted SD's, including SD's below the lower limit of quantification.

## Conclusions

In the framework of precision pharmacotherapy, establishing the AEE of assayed drugs is the responsibility of the therapeutic drug monitoring service. This permits optimal dosages by providing the correct weighting factor of assay results in the development of population and individual pharmacokinetic models.

## Introduction

It is increasingly accepted that the utility of therapeutic drug monitoring (TDM) can be improved significantly by integrating the assayed drug concentrations into population and individualized clinical pharmacokinetic models. This can be optimized by using correct weighting of the data in making such models [1–4], enabling the maximally precise calculation of individual patient dosages using precision pharmacotherapy software. Because the correct weighting factor is proportional to 1/variance, all such models require reporting the standard deviation (SD) values [not the coefficients of variation (CV%)] corresponding to the measured drug levels. (Fig 1) [5,6].

Increasing evidence suggests that most bioanalytical assays are heteroscedastic, i.e. the SD varies in some way with the measured concentration [7,8]. The relationship between SD and analyte concentration can be estimated quantitatively for each level measured by the analytical method. This approach was first proposed by Ekins et al for an assay of aldosterone, revisited by Sadler, and proposed as the experimentally quantified error term in both population pharmacokinetic (PK) modeling and Bayesian posterior models in individual patients by Jelliffe and Tahani [9–11]. The clinical value of quantifying the assay error as a function of the drug concentration has resulted in its incorporation into precision pharmacotherapy software suites employing nonparametric pharmacokinetic modeling, such as Pmetrics™ and BestDose™ [12,13].

Based on studies conducted using automated clinical analyzers in the 1990's, unweighted second degree polynomial regression was proposed to characterize the assay precision profile, established by analyzing several replicates of real patient specimens [6,11]. Recent advances show that the future of TDM is strongly linked to liquid chromatography-tandem mass spectrometry (LC-MS/MS) [14]. As shown in practice on a voriconazole assay and on theoretical grounds, bioanalytical LC-MS/MS methods can be characterized by linear error patterns [6,8].

In an effort to increase performance and decrease laboratory turnaround times and operational costs, automated clinical assays, including drug analysis, are typically performed without repeats. Therefore, the SD of the drug assay result, reported for further PK calculations, needs to be estimated. For this purpose, assay error equations (AEE's) can be easily used. AEE's are developed systematically, from zero concentration throughout the entire working range of the assay, by performing regression on the series of concentration-SD data pairs (i.e. the precision profile) with multiple measurements done over the widest measurable concentration range

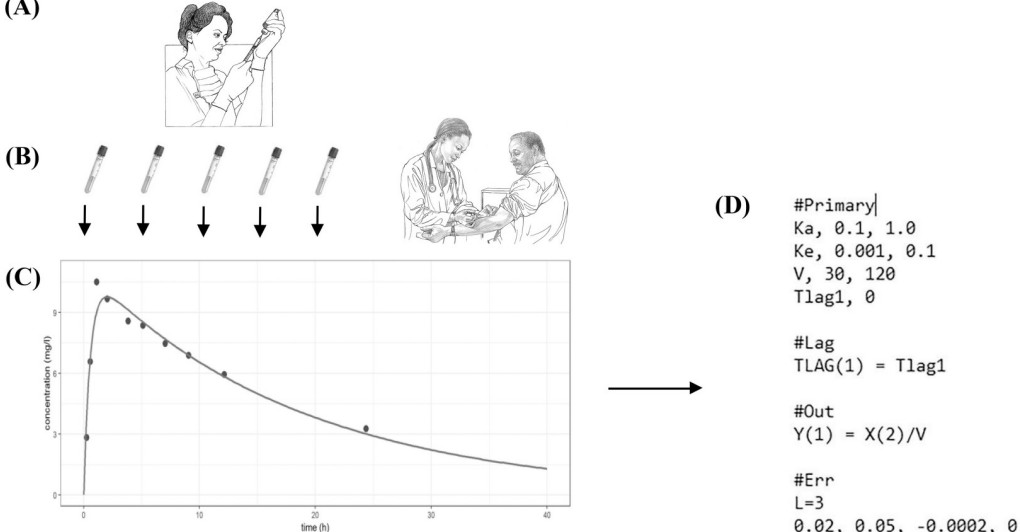

**Fig 1. An overview of how therapeutic drug monitoring (TDM) results can be integrated into clinical population and individual pharmacokinetic models on which individualized drug therapy is based.** After the drug is administered (A), blood is drawn at specific times (B) and submitted for analysis to obtain time-concentration data with the aim of determining the subject's individual pharmacokinetic parameters. This can be done using, for instance, either parametric or nonparametric Bayesian analysis (C). An illustrative input pharmacokinetic model file of Pmetrics™, a nonparametric population modeling software suite [12], is shown (D). The coefficients of the assay error equation are included in the computer script under "#Err". Ka, Ke, V and Tlag are notations for pharmacokinetic parameters. Line art images are uncopyrighted and were made available by the National Institute of Diabetes and Digestive and Kidney Diseases, National Institutes of Health, United States at https://www.niddk.nih.gov/news/media-library.

and enrolling several independent serum samples fortified with the analyte(s). Furthermore, an appropriate number of independent experiments is performed to include the inter-day variations of laboratory operation.

Although it has been advocated that using AEE's may extend the clinical utility of bioanalytical methods, to our knowledge, no systematic analysis has been published on the methodology of AEE development [15].

An appropriate AEE will significantly optimize the precision of parameter estimates in population and individual patient pharmacokinetic models. Currently, little attention is given to this issue, and CV% is still regarded generally as the primary measure of assay precision. Our aim is to present an approach to develop AEE's which allows the most precise calculation of the SD's for each result obtained in routine clinical LC-MS/MS drug assays. For this purpose we studied precision profiles using an in-house LC-MS/MS method developed for the analysis of carbamazepine (CBZ), fluconazole (FLU), lamotrigine (LAM) and levetiracetam (LEV) [2,16,17].

## Materials and methods

### Chemicals and solutions

CBZ, FLU, LAM, LEV (all >99.5% pure), and their isotopically labeled analogs fluconazole-$^2$H$_4$ 98.0%, lamotrigine-$^{13}$C$_7$,$^{15}$N 99.0%, and levetiracetam-$^2$H$_6$ 99.5% were purchased from Alsachim SAS (Ilkirch-Graffenstaden, France). Carbamazepine-$^2$H$_{10}$ 0.1 mg/mL in methanol solution (CBZ-ISS) was a certified reference material bought from Sigma-Aldrich Hungary Kft (Budapest, Hungary). LC-MS grade acetonitrile, methanol and water were obtained from Molar Chemicals Hungary Kft (Halásztelek, Hungary).

Stock solutions of the analytes were prepared in methanol and stored at -75 ˚C until use. The first series of stock solutions contained CBZ (CBZ-SS1), FLU (FLU-SS1), LAM (LAM-SS1) or LEV (LEV-SS1) at 4.006, 4.948, 4.080 or 3.974 mg/mL, respectively. The second series of stock solutions contained the analytes at 2.017 (CBZ-SS2), 2.014 (FLU-SS2), 1.987 (LAM-SS2) and 2.015 (LEV-SS2) mg/mL, respectively. The stock solutions of the internal standards fluconazole-$^2$H$_4$ (FLC-ISS), lamotrigine $^{13}$C$_7$ (LAM-ISS) and levetiracetam $^2$H$_6$ (LEV-ISS), prepared in methanol, contained the substances at 1.00 mg/mL. The internal standard solution (IS) contained carbamazepine-$^2$H$_{10}$ at 0.2 μg/mL as well as fluconazole-$^2$H$_4$, lamotrigine-$^{13}$C$_7$,$^{15}$N, and levetiracetam-$^2$H$_6$ at 2 μg/mL in acetonitrile.

The analyte solution used in the carry-over experiment was prepared by spiking CBZ-SS1, FLU-SS1, LAM-SS1 and LEV-SS1, 5 μL each, to 980 μL methanol, and by adding 10 μL of the resulting mixture to 990 μL methanol-water (3:1, v/v).

Further solutions were used in the matrix factor study. The high-level matrix factor spiking solution (MF-H) was prepared by diluting CBZ-SS1, FLC-SS1, LAM-SS1 and LEV-SS1, 25 μL each, to 500 μL with methanol. The low-level matrix factor spiking solution (MF-L) was prepared by diluting 10 μL MF-H with 990 μL methanol. The internal standard spiking solution (MF-IS) was prepared by diluting CBZ-ISS, FLC-ISS, LAM-ISS and LEV-ISS, 90 μL each, to 1000 μL with methanol.

## Serum specimens

All of our procedures were in agreement with the Declaration of Helsinki. There was no interaction or intervention with human subjects in the framework of our study, and only leftover patient-related physical material was used. All employed serum specimens were irreversibly de-identified prior to making any manipulation or measurements related to our study on these specimens. Identifiable private information was not accessed by the authors. Due to these considerations and in line with effective regulations, no approval was sought from the Institutional Ethics Committee of Semmelweis University.

In this work, the word 'specimen' refers to unmanipulated blood or serum separated therefrom. 'Sample' is used for serum prepared, or manipulated by spiking the analytes, and then prepared for analysis. 'Sample set' is a batch of samples prepared from serum specimens, each of which had been obtained from a different individual. Human serum was obtained after blood had been drawn into 6-mL red-top native, serum-separator tubes in a standard phlebotomy process from in- and outpatients at various Departments of Semmelweis University (Budapest, Hungary), sent to the Central Laboratory (Pest), Department of Laboratory Medicine, Semmelweis University for routine diagnostic tests unrelated to our study, and by separating the serum by centrifugation (5366 x g, 10 min, cooled to +4 ˚C). The specimens employed in this study were leftover material which had undergone all ordered tests and were stored separately at 2–8 ˚C, allocated for disposal. All information displayed on the tubes and allowing patient identification were removed irreversibly at the Central Laboratory (Pest), and the tubes were given a numerical identifier for use in our study.

Candidate specimens were tested for hyperbilirubinemia, hyperlipidemia and hemolysis by establishing the lipemia-icterus-hemolysis index (LIH index, lipemic index) using a Beckman-Coulter clinical analyzer (Beckman Coulter Hungary, Budapest, Hungary) at Central Laboratory (Pest). Specimens with a positive LIH index were inspected visually for making judgement on which type of interference was dominant. Subsequently, specimens were frozen at -18 ˚C and transported in a frozen state to the Laboratory of Mass Spectrometry and Separation Technology, Department of Laboratory Medicine, Semmelweis University, for processing in the

framework of this study. The absence of the analytes from the specimens was verified in preliminary experiments using the methodology described below.

## Experimental design

This study consisted of 3 experiments conducted on separate sets of human serum specimens of at least 2.0 mL in volume, all collected from different patients. Spiked specimens (i.e. samples) were stored at -75 ˚C until analysis. The specimens were not pooled before spiking. In both experiments 1 and 2, specimens with a positive LIH index (hyperbilirubinemic, hemolytic and lipemic, 6 of each) were selected, in addition to 6 specimens without these conditions (termed as "normal"). The total of 24 specimens were spiked with the analytes at 4 and 6 concentration levels in each (i.e. spiking levels) in experiments 1 and 2, respectively. Experiment 3 was carried out by spiking 24"normal" serum specimens to reach 10 different concentration levels for each analyte (Fig 2).

In each experiment, various regression equations were applied to characterize the precision profiles obtained on sets of 6 ("small" sample sets), as well as of 20 spiked serum specimens ("large" sample sets). To evaluate the results obtained in all possible combinations of the samples, the combination formula "N choose R": $\binom{N}{R} = \frac{N!}{N!(N-R!)}$, was employed, where N is the total number of samples included in an experimental set (N = 24 at each spiking level) and R is the number of samples in which the drug concentrations were evaluated to obtain the precision profiles (R = 6 and R = 20 for "small" and for "large" sample sets, respectively, at each spiking level). Consequently, in each experiment and for each type of regression, 134596 (when R = 6) and 10626 (when R = 20) sets of coefficients were obtained. The calculations were performed using the computer scripts provided in S3 File.

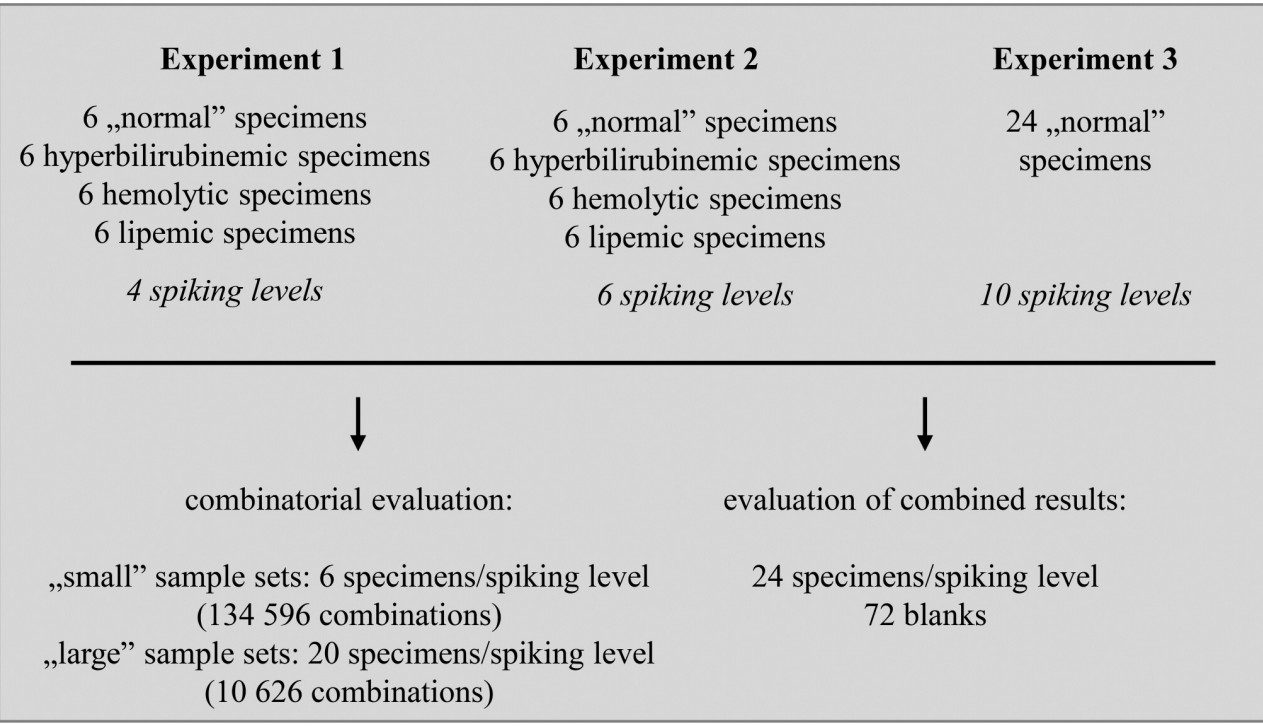

**Fig 2. Presentation of the experimental design.** 3 experiments were performed. The impact of sample size, the number of spiking levels and common interferences was assessed on "small" and"large" sample sets containing 6 and 20 samples, respectively, using a combinatorial approach. The final assay error equations were established by using all of the results obtained in the 3 experiments (N = 24 specimens/spiking level, M = 20 spiking levels+blank).

The final AEE's, intended for use in developing pharmacokinetic models, were established by performing regression on the precision profiles established using each and every assay result obtained in the 3 independent experiments (N = 24 specimens at each spiking level, M = 20 spiking levels, Fig 2).

### Preparation of serum calibrator and spiked independent samples

Separate calibrator series, consisting of 6 levels each, were prepared freshly for each experiment. Pooled serum previously screened for the absence of the investigated substances was used. S1 and S2 Tables present details of the preparation and the analyte content of the calibrators and the spiked independent samples, respectively.

### Sample preparation

Spiking of the analytes was performed by adding a small volume of the methanol solution of the analytes (i.e. the spiking volume) to the specimens, corresponding to ≤9.5% of the specimen volume.

Sample preparation was identical for all 4 analytes. 25 μL serum and 200 μL IS were pipetted onto a Phenomenex Impact™ 96-well protein precipitation plate (Gen-Lab Kft, Budapest, Hungary), which was shaken (1100 rpm, 10 min) on an Allsheng TMS-200 thermo shaker (Labex Kft, Budapest, Hungary). The supernatant was forced into a Brand® 96-well deep well collection plate (Merck Kft, Budapest, Hungary) using a Phenomenex® Presston 100 positive pressure manifold (Gen-Lab Kft, Budapest, Hungary) and nitrogen gas (5.0 purity, Messer Hungarogáz Kft, Budapest, Hungary). Then, a 25 μL-aliquot of the supernatant was transferred into another collection plate and 475 μL water-methanol (3:1, v/v) mixture was added. The entire procedure was performed at room temperature.

### Analytical method

Analysis was performed using a Shimadzu Nexera X2 ultra-performance liquid chromatograph consisting of two LC-30 pumps, a SIL-30AC autosampling unit and a CTO-20AC column thermostat coupled to a Shimadzu LCMS-8060 mass spectrometer equipped with an electrospray ionization source and operated with positive polarity. Integrated system control and data acquisition were conducted by the Shimadzu LabSolutions v5.89 software (Simkon Kft, Budapest, Hungary).

Chromatography was performed using a Phenomenex Kinetex XB-C18 50x2.1 mm (1.7 μm) analytical column kept at 40 ˚C and protected by a Phenomenex® SecurityGuard Ultra C18 2.1x5 mm cartridge (Gen-Lab Kft, Budapest, Hungary). The mobile phases were water (A) and methanol (B), both containing 0.1% formic acid. The gradient program was 0 min: 30% B, 0.50 min: 30% B, 2.00 min: 90% B, 3.00 min: 90% B, 3.01 min: 30% B. The run time was 4.00 min. The sampler tray was cooled to 15 ˚C. 2-μL aliquots were injected.

The general mass spectrometer settings were as follows. Nebulizer gas: 3 L/min, heating gas: 12 L/min, drying gas: 6 L/min, interface temperature: 300 ˚C, desolvation line temperature: 250 ˚C, heat block temperature: 400 ˚C. The dwell time of each ion transition was 50 ms. Analyte-specific detection settings are shown in Table 1. Each spiked serum specimen was tested in a single measurement.

### Evaluation of analytical results

The compounds of interest were identified by their ion transitions and retention times. Peak area ratios of the target ion transitions of analytes and internal standards were used for quantitative analysis. The 6-point calibration model was linear with $1/x^2$ weights, performed at the

**Table 1. Mass spectrometry settings applied for the quantitation of carbamazepine, fluconazole, lamotrigine and levetiracetam.**

| Analyte | | Precursor ion | Product ion | Q1 bias (V) | Collision energy (V) | Q3 bias (V) |
|---|---|---|---|---|---|---|
| Carbamazepine (retention time: 2.6 min) | target | 236.9 | 193.1 | -11 | -34 | -20 |
| | qualifier | 237.0 | 179.1 | -14 | -36 | -19 |
| | internal standard target | 246.9 | 203.1 | -11 | -34 | -20 |
| | internal standard qualifier | 246.9 | 174.1 | -14 | -36 | -19 |
| Fluconazole (retention time: 1.9 min) | target | 306.9 | 238.1 | -14 | -17 | -25 |
| | qualifier | 306.9 | 169.1 | -15 | -23 | -18 |
| | internal standard target | 310.9 | 242.1 | -14 | -17 | -25 |
| | internal standard qualifier | 310.9 | 173.1 | -15 | -23 | -18 |
| Lamotrigine (retention time: 1.4 min) | target | 255.9 | 211.0 | -25 | -26 | -22 |
| | qualifier | 255.9 | 145.0 | -23 | -39 | -22 |
| | internal standard target | 263.9 | 218.0 | -25 | -26 | -22 |
| | internal standard qualifier | 263.9 | 151.0 | -23 | -39 | -22 |
| Levetiracetam (retention time: 0.9 min) | target | 171.0 | 126.1 | -18 | -15 | -13 |
| | qualifier | 171.0 | 154.1 | -25 | -12 | -16 |
| | internal standard target | 177.0 | 132.1 | -18 | -15 | -13 |
| | internal standard qualifier | 177.0 | 160.1 | -25 | -12 | -16 |

beginning of each batch run. In experiments 2 and 3, concentrations lower than the calibrated range were evaluated by single point calibration using the lowest level calibrator (0.403, 0.403, 0.397 and 0.403 μg/mL for CBZ, FLU, LAM and LEV, respectively).

## Method validation

Method validation was carried out according to the respective guideline on bioanalytical method validation issued by the European Medicines Agency [18]. Sample carry-over with respect to the signals of the analytes, the performance of the calibration curves, matrix factors, as well as accuracy and precision (the latter as the CV%) were evaluated. Since the aim of this work was to establish precision profiles using specimens spiked immediately before conducting the experiments, storage or freeze-thaw stability studies were not performed.

Sample carry-over was tested by injecting 2 μL of water-methanol (3:1, v/v) solutions containing CBZ, FLU, LAM and LEV at 200, 247, 204 and 198 ng/mL, respectively, 5 times, and by evaluating the areas of analyte and internal standard peaks potentially appearing in the ion chromatograms of blank solvents run in between.

Matrix effect was tested by evaluating matrix factors and the internal-standard corrected matrix factors in "normal", hyperbilirubinemic, hemolytic and lipemic specimens, 6 of each. Deproteinization was accomplished by diluting 25 μL blank specimen with 200 μL acetonitrile as described in the 'Sample preparation' section. 190 μL supernatant was transferred to a 2-mL Eppendorf tube and was spiked with 5 μL MF-H or MF-L, as well as with 5 μL MF-IS. 25 μL of the spiked supernatant was diluted with 475 μL water-methanol (3:1, v/v). Reference solutions were prepared by adding 5 μL MF-H or MF-L, as well as 5 μL MF-IS, to 200 μL acetonitrile-water (8:1, v/v) and diluting a 25-μL aliquot of this mixture with 475 μL water-methanol (3:1, v/v). The analyte content of the spiked supernatants corresponded to CBZ, FLU, LAM and LEV serum concentrations of 0.451, 0.557, 0.459 and 0.447 μg/mL in the low, as well as 45.1, 55.7, 45.9 and 44.7 μg/mL in the high level samples, respectively.

In this paper, the term „lower limit of quantification (LLOQ)" refers to the lowest concentration level of each calibration model, as set forth in [18] and not to the lowest level judged as quantifiable based on signal-to-noise ratios.

## Development of AEE's by applying linear and nonlinear regression to the precision profiles

The performance of the AEE's was compared in terms of the differences between the experimentally determined SD's and the SD's estimated using (a) unweighted (ordinary) linear, (b) second- and (c) third-order least squares, (d) $1/x^2$-weighted linear least squares, and Theil's regression, with (e) and without (f) using the Siegel estimator [19,20]. Theil's regression is a robust nonparametric median-based linear regression algorithm which uses Eqs (1) and (2) [21]:

$$slope = median\left[\frac{(SD_j - SD_i)}{(concentration_j - concentration_i)}\right] \tag{1}$$

$$intercept = median\left(SD_i - slope \cdot concentration_i\right) \tag{2}$$

with M spiking levels. The estimated slope was obtained by first calculating the slopes of the lines connecting each i-th SD to all of the j-th SD's, where j>i, i = 1...(M-1), and j = 2...M, and, second, by taking their median. Subsequently, the intercepts were calculated [again, i = 1...(M-1)], and their median was determined to obtain the estimate.

The Siegel estimator is a modification of Theil's regression which uses repeated medians as follows:

$$\hat{a} = median\left[\begin{array}{c} median \\ i \neq j \end{array} \frac{(SD_j - SD_i)}{(concentration_j - concentration_i)}\right] \tag{3}$$

$$\hat{c} = median\left[\begin{array}{c} median \\ i \neq j \end{array} \frac{(concentration_j SD_i - concentration_i SD_j)}{(concentration_j - concentration_i)}\right] \tag{4}$$

where $\hat{a}$ is the estimated slope and $\hat{c}$ is the estimated intercept. One of the differences from Theil's original formula here is that the estimate of the slope is obtained by calculating the individual slopes of the lines connecting each i-th SD to all others (j<i as well as j>i, i = 1...M), recording the median of each i-th slope [median(i≠j)], and, after obtaining all individual medians, determining the overall median, i.e. the overall estimate of the slope. The calculation of the intercept is similarly performed in Eq (4). Theil's algorithm is frequently employed with modifications of Sen (referred to as Theil-Sen regression [22]). However, in this study Theil's original approach was applied.

The performance of the various regression approaches was evaluated by determining the sums of squared residuals normalized to those predicted by the regression algorithm (NSSR). NSSR is an unbiased estimator which retains the widely employed concept of judging the goodness of fit by calculating the residuals [Eq (5)]:

$$NSSR = \sum_i^M \frac{(SD_{observed,i} - SD_{predicted,i})^2}{SD_{predicted,i}^2} \tag{5}$$

Computer scripts were written by the authors in the R environment (version 3.5.1) to perform the calculation of SD's, to generate and plot precision profiles, and to perform, as well as to evaluate and plot the results of the regression using algorithms (a)-(f) [23]. Theil's regression [algorithms (e) and (f)] was performed using the mblm() function of the 'mblm' package. Weighted and unweighted linear and nonlinear least squares were fitted using the lm() function of the 'stats' package. In each experiment and for each analyte, the medians and ranges of the regression coefficients were calculated. Plots were created using the 'ggplot2' package. The

input files, created in comma-separated value (.csv) format using Microsoft Excel, contained the analytical results (nominal and measured concentrations obtained for each specimen at each spiking level, S1 and S2 Files). An interactive website is under construction to make the procedure presented here available to interested professionals.

## Results

### Method validation

The analytical method described here allowed the selective and specific detection of the analytes in human serum specimens with conventional detection limits of 0.144 pg, 0.216 pg, 0.213 pg and 0.216 pg for CBZ, FLU, LAM and LEV, respectively. The retention times were 2.6, 1.9, 1.4 and 0.9 min, respectively. No sample carry-over was observed. Typical ion chromatograms and representative calibration plots are shown in Fig 3.

The performance of the calibration models used in the 3 independent experiments are shown in Table 2. The intercepts were negligible and the differences of their means from zero was not significant, as evaluated by performing a one-sample Welch test (p = 0.4099,

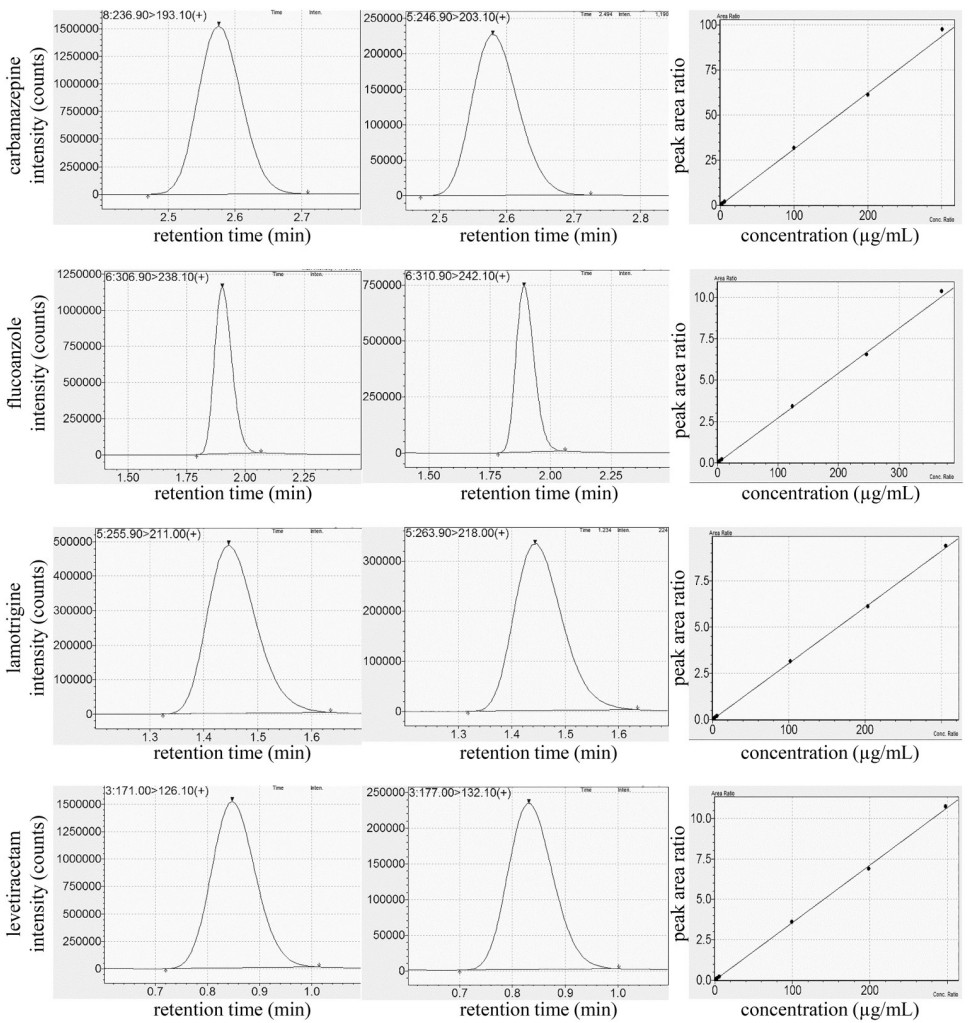

**Fig 3. Representative analytical data.** Ion chromatograms of the analytes (left) and the internal standards (middle), and representative calibration curves (right) obtained in the experiments.

**Table 2. Performance of the calibration curves employed in this study.**

| Analyte | Characteristic | Experiment 1 | Experiment 2 | Experiment 3 |
|---|---|---|---|---|
| CBZ | Slope | 0.312 | 0.306 | 0.316 |
| | Intercept | -0.0742 | -0.00177 | 0.000111 |
| | Determination coefficient | 0.998 | 0.999 | 0.991 |
| | Accuracies of back-calculated concentrations | 95.8–105% | 97.2–104% | 92.2–115% |
| FLC | Slope | 0.0288 | 0.0271 | 0.0327 |
| | Intercept | -0.00751 | -0.000101 | -0.000976 |
| | Determination coefficient | 0.999 | 0.999 | 0.998 |
| | Accuracies of back-calculated concentrations | 96.2–103% | 98.0–103% | 96.6–108% |
| LAM | Slope | 0.0260 | 0.0303 | 0.0338 |
| | Intercept | -0.00870 | -0.000264 | -0.00200 |
| | Determination coefficient | 0.9994 | 0.9998 | 0.9973 |
| | Accuracies of back-calculated concentrations | 96.9–103% | 98.6–102% | 94.8–108% |
| LEV | Slope | 0.0320 | 0.0356 | 0.0349 |
| | Intercept | -0.00648 | -0.0000157 | -0.0000279 |
| | Determination coefficient | 0.9992 | 0.9996 | 0.9963 |
| | Accuracies of back-calculated concentrations | 97.1–104% | 97.4–102% | 95.4–110% |

$p = 0.3454$, $p = 0.2912$ and $p = 0.4188$, respectively). All of the back-calculated concentrations of the calibration standards were accepted by applying the recommendations of [18].

The accuracy of CBZ, FLU, LAM and LEV concentrations measured in the spiked serum samples ranged between 89.2–133%, 93.1–107%, 80.3–109% and 89.8–109% with CV% ranges of 2.51–17.7%, 2.96–16.6%, 2.19–17.0% and 1.79–11.4%, respectively (Table 3).

The results of the matrix factor experiments did not show the suppression or enhancement of the ionization of the analytes or the internal standards in the "normal", hyperbilirubinemic, hemolytic or lipemic samples. The matrix factors ranged between 98.1–111%, and the internal standard-corrected matrix factors were 98.9–114% (S3 Table).

## Combinatorial evaluation of precision profiles in experiments 1–3

The medians and ranges of the linear coefficients of the AEE's established by performing regression on SD's obtained in sets of R = 6 or R = 20 serum samples are shown in Table 4. These medians (but not the ranges) showed good agreement regardless of the regression algorithm or the experimental setup applied. In addition, sample size (R) caused negligible differences between the medians of the linear coefficients obtained using the same regression algorithm. The medians of the SD's obtained in experiment 3 conducted with „normal" serum specimens were lower than those obtained in experiments 1 and 2.

When linear regression [algorithms (a), (d), (e) or (f)] was applied, the ratios of the highest/lowest linear coefficients (Table 4A, 'high/low' values) obtained for CBZ, FLU, LAM and LEV were <18.0, <18.4, <25.2 and <13.6, respectively, in the "small", and <1.96, <1.95, <10.8 and <1.81, respectively, in the "large" sample sets. These data show that the variability of the linear coefficients (slopes) of the linear assay error equations dropped by approximately ten-fold when the number of independent samples assayed at each spiking level was increased from 6 to 20. In one experimental setup (experiment 3, M = 10, R = 6), $1/x^2$-weighted linear least squares regression [algorithm (d)] performed on LAM yielded negative linear coefficients in 24.4% of the sample combinations, and, for the "large" sample set of the same experiment (M = 10, R = 20), a highest/lowest linear coefficient ratio of 10.8, which exceeded dramatically those obtained in all other experimental setups with "large" sample sets (1.16–1.59).

**Table 3. Within-run accuracy and relative precision (expressed as the coefficient of variation) obtained for the analytes in spiked independent serum samples in the conducted experiments.**

| Experiment number | Spiking level | Carbamazepine | | | | Fluconazole | | | |
|---|---|---|---|---|---|---|---|---|---|
| | | Nominal analyte concentration (µg/mL) | Mean measured concentration (µg/mL) | Accuracy (%) | CV (%) | Nominal analyte concentration (µg/mL) | Mean measured concentration (µg/mL) | Accuracy (%) | CV (%) |
| 1 | 1 | 3.02 | 2.71 | 89.8 | 9.12 | 1.24 | 1.20 | 96.2 | 6.57 |
| | 2 | 15.1 | 13.5 | 89.2 | 6.52 | 6.21 | 5.97 | 96.2 | 5.47 |
| | 3 | 60.3 | 54.4 | 90.2 | 7.06 | 24.8 | 24.0 | 96.5 | 7.15 |
| | 4 | 181 | 164 | 90.6 | 6.94 | 74.5 | 71.9 | 96.5 | 4.72 |
| 2 | 1 | 0.209 | 0.194 | 92.7 | 8.95 | 0.102 | 0.107 | 106 | 10.7 |
| | 2 | 0.419 | 0.407 | 97.2 | 10.3 | 0.203 | 0.209 | 103 | 13.3 |
| | 3 | 2.09 | 2.00 | 95.7 | 6.84 | 1.02 | 0.969 | 95.4 | 6.91 |
| | 4 | 8.38 | 8.96 | 107 | 7.28 | 4.07 | 4.19 | 103 | 5.40 |
| | 5 | 16.8 | 18.4 | 110 | 6.06 | 8.13 | 8.58 | 105 | 5.30 |
| | 6 | 33.5 | 37.5 | 112 | 5.09 | 16.3 | 17.4 | 107 | 5.01 |
| 3 | 1 | 0.0122 | 0.0159 | 133 | 17.7 | 0.0122 | 0.0118 | 97.1 | 16.6 |
| | 2 | 0.0243 | 0.0293 | 120 | 8.16 | 0.0243 | 0.0226 | 93.1 | 8.82 |
| | 3 | 0.0486 | 0.0588 | 118 | 8.53 | 0.0486 | 0.0472 | 97.1 | 7.55 |
| | 4 | 0.0972 | 0.116 | 119 | 7.68 | 0.0972 | 0.101 | 104 | 8.04 |
| | 5 | 0.243 | 0.253 | 104 | 4.45 | 0.243 | 0.239 | 98.3 | 4.42 |
| | 6 | 0.810 | 0.836 | 103 | 2.51 | 0.810 | 0.810 | 100 | 2.96 |
| | 7 | 2.43 | 2.52 | 104 | 2.57 | 2.43 | 2.44 | 100 | 3.64 |
| | 8 | 4.21 | 4.15 | 98.6 | 3.72 | 4.21 | 4.16 | 98.9 | 3.67 |
| | 9 | 12.2 | 11.8 | 96.6 | 3.61 | 12.2 | 12.3 | 101 | 2.97 |
| | 10 | 24.3 | 24.2 | 99.7 | 4.06 | 24.3 | 24.9 | 102 | 3.82 |
| Experiment number | Spiking level | Lamotrigine | | | | Levetiracetam | | | |
| | | Nominal analyte concentration (µg/mL) | Mean measured concentration (µg/mL) | Accuracy (%) | CV (%) | Nominal analyte concentration (µg/mL) | Mean measured concentration (µg/mL) | Accuracy (%) | CV (%) |
| 1 | 1 | 3.07 | 2.91 | 94.6 | 7.43 | 2.49 | 2.24 | 89.8 | 8.21 |
| | 2 | 15.4 | 14.6 | 94.7 | 5.97 | 12.5 | 10.6 | 93.3 | 5.87 |
| | 3 | 61.4 | 58.1 | 94.5 | 7.36 | 49.9 | 46.8 | 93.8 | 6.14 |
| | 4 | 184 | 174 | 94.6 | 5.15 | 150 | 141 | 94.1 | 5.88 |
| 2 | 1 | 0.251 | 0.202 | 80.3 | 6.76 | 0.208 | 0.203 | 97.8 | 8.36 |
| | 2 | 0.503 | 0.416 | 82.8 | 8.76 | 0.416 | 0.408 | 98.3 | 8.25 |
| | 3 | 2.51 | 2.06 | 82.0 | 4.27 | 2.08 | 1.94 | 93.4 | 6.46 |
| | 4 | 10.1 | 9.33 | 92.7 | 5.29 | 8.31 | 8.27 | 99.5 | 4.84 |
| | 5 | 20.1 | 19.9 | 98.8 | 4.55 | 16.6 | 16.7 | 101 | 4.48 |
| | 6 | 40.2 | 41.5 | 103 | 4.11 | 33.2 | 33.1 | 99.4 | 4.35 |
| 3 | 1 | 0.0120 | 0.0123 | 103 | 17.0 | 0.0486 | 0.0528 | 109 | 11.4 |
| | 2 | 0.0239 | 0.0233 | 97.3 | 10.8 | 0.0971 | 0.0997 | 103 | 8.33 |
| | 3 | 0.0478 | 0.0479 | 99.8 | 10.0 | 0.194 | 0.198 | 102 | 7.31 |
| | 4 | 0.0956 | 0.104 | 109 | 8.68 | 0.388 | 0.406 | 105 | 7.11 |
| | 5 | 0.239 | 0.240 | 100 | 3.82 | 0.971 | 1.00 | 103 | 3.81 |
| | 6 | 0.797 | 0.807 | 101 | 2.87 | 3.24 | 3.27 | 101 | 2.22 |
| | 7 | 2.39 | 2.44 | 102 | 2.24 | 9.71 | 9.61 | 98.9 | 2.63 |
| | 8 | 4.14 | 4.16 | 101 | 2.50 | 16.8 | 16.46 | 98.0 | 2.25 |
| | 9 | 12.0 | 12.3 | 103 | 2.19 | 48.5 | 48.4 | 99.8 | 1.79 |
| | 10 | 23.9 | 24.7 | 103 | 3.42 | 97.1 | 91.4 | 94.2 | 5.88 |

CV%: coefficient of variation.

**Table 4. Medians and ranges of linear and constant coefficients of carbamazepine, fluconazole, lamotrigine and levetiracetam assay error equations obtained by developing precision profiles for all combinations of 6 or 20 samples in the three experiments conducted.** (a) unweighted linear least squares regression, (b) unweighted $2^{nd}$-order least squares polynomial regression, (c) unweighted $3^{rd}$-order least squares polynomial regression, (d) $1/x^2$-weighted least squares polynomial regression, (e) Theil's regression, (f) Theil's regression using the Siegel estimator. $3^{rd}$-order least squares regression was not applied in experiment 1 due to the small number of spiking levels (M = 4). (A) Linear coefficients. High/low, ratio of the highest and the lowest linear coefficients obtained in the combinatorial calculation of SD's and, subsequently, assay error equations in each experimental setup. ND, not determined. (B) Constant coefficients (intercepts). ND, not determined. NNI, percentage proportions of non-negative intercepts obtained in the combinatorial calculation of SD's and, subsequently, assay error equations in each experimental setup.

(A)

| regression algorithm | spiking levels | samples/ level | Carbamazepine median (range) | high/low | Fluconazole median (range) | high/low | Lamotrigine median (range) | high/low | levetiracetam median (range) | high/low |
|---|---|---|---|---|---|---|---|---|---|---|
| (a) | 4 | 6 | 0.0636 (0.0089–0.1353) | 15.1 | 0.0390 (0.0065–0.0875) | 13.4 | 0.0431 (0.0038–0.0967) | 25.2 | 0.0564 (0.0080–0.1090) | 13.6 |
| | 4 | 20 | 0.0703 (0.0392–0.0763) | 1.95 | 0.0492 (0.0266–0.0522) | 1.96 | 0.0542 (0.0331–0.0576) | 1.74 | 0.0605 (0.0368–0.06466) | 1.76 |
| | 6 | 6 | 0.0473 (0.0047–0.0846) | 18.0 | 0.0454 (0.0073–0.0856) | 11.8 | 0.0350 (0.0070–0.0758) | 10.8 | 0.0425 (0.0059–0.0755) | 12.8 |
| | 6 | 20 | 0.0528 (0.0335–0.0564) | 1.69 | 0.0513 (0.0277–0.0544) | 1.96 | 0.0426 (0.0256–0.0453) | 1.77 | 0.0436 (0.0259–0.0468) | 1.81 |
| | 10 | 6 | 0.0383 (0.0050–0.0680) | 13.7 | 0.0355 (0.0033–0.0613) | 18.4 | 0.0317 (0.0045–0.0521) | 11.7 | 0.0518 (0.0085–0.0784) | 9.22 |
| | 10 | 20 | 0.0403 (0.0295–0.0439) | 1.49 | 0.0373 (0.0240–0.0397) | 1.65 | 0.0323 (0.0246–0.0352) | 1.43 | 0.0524 (0.0431–0.0566) | 1.31 |
| (b) | 4 | 6 | 0.0633 (-0.0269–0.1847) | ND | 0.0756 (-0.0136–0.1896) | ND | 0.0818 (-0.0163–0.2146) | ND | 0.0641 (-0.0332–0.1123) | ND |
| | 4 | 20 | 0.0731 (0.0238–0.0896) | 3.76 | 0.0888 (0.0418–0.0989) | 2.36 | 0.0989 (0.0416–0.1111) | 2.67 | 0.0626 (0.0422–0.0712) | 1.69 |
| | 6 | 6 | 0.0615 (-0.0219–0.1307) | ND | 0.0551 (-0.0116–0.1090) | ND | 0.0506 (-0.0166–0.0999) | ND | 0.0453 (-0.0236–0.1001) | ND |
| | 6 | 20 | 0.0641 (0.0413–0.0770) | 1.86 | 0.0548 (0.0417–0.0673) | 1.61 | 0.0506 (0.0380–0.0623) | 1.64 | 0.0453 (0.0310–0.0587) | 1.90 |
| | 10 | 6 | 0.0304 (-0.0203–0.0948) | ND | 0.0173 (-0.0207–0.0690) | ND | 0.0099 (-0.0203–0.0444) | ND | -0.0139 (-0.0545–0.0363) | ND |
| | 10 | 20 | 0.0322 (0.0126–0.0445) | 3.53 | 0.0260 (0.0046–0.0318) | 6.87 | 0.0126 (0.0028–0.0196) | 6.91 | -0.0139 (-0.0230 – -0.0042) | ND |
| (c) | 4 | 6 | | | | | Not evaluated | | | |
| | 4 | 20 | | | | | Not evaluated | | | |
| | 6 | 6 | 0.0462 (-0.1053–0.1705) | ND | 0.0507 (-0.0602–0.1676) | ND | 0.0613 (-0.0678–0.1676) | ND | 0.0541 (-0.0615–0.1572) | ND |
| | 6 | 20 | 0.04579 (0.0084–0.0787) | 9.36 | 0.0549 (0.0129–0.0726) | 5.63 | 0.0605 (0.0238–0.0883) | 3.71 | 0.0580 (0.0226–0.0750) | 3.31 |
| | 10 | 6 | 0.0301 (-0.0209–0.0823) | ND | 0.0364 (-0.0020–0.0784) | ND | 0.0260 (-0.0088–0.0572) | ND | 0.0334 (0.0007–0.0597) | 82.5 |
| | 10 | 20 | 0.0324 (0.0170–0.0422) | 2.49 | 0.0431 (0.0264–0.0490) | 1.85 | 0.0268 (0.0169–0.0344) | 2.04 | 0.0337 (0.0250–0.0413) | 1.66 |
| (d) | 4 | 6 | 0.0702 (0.0334–0.1115) | 3.34 | 0.0551 (0.0167–0.1034) | 6.21 | 0.0612 (0.0198–0.1119) | 5.64 | 0.0611 (0.0029–0.1001) | 3.50 |
| | 4 | 20 | 0.0733 (0.0611–0.0793) | 1.30 | 0.0585 (0.0404–0.0642) | 1.59 | 0.0650 (0.0480–0.0711) | 1.48 | 0.0654 (0.0522–0.0669) | 1.28 |
| | 6 | 6 | 0.0657 (0.0291–0.0979) | 3.36 | 0.0686 (0.0232–0.1167) | 5.04 | 0.0516 (0.0268–0.0805) | 3.01 | 0.0563 (0.0212–0.0842) | 3.98 |
| | 6 | 20 | 0.0687 (0.0598–0.0733) | 1.23 | 0.0742 (0.0524–0.0803) | 1.53 | 0.0554 (0.0469–0.0589) | 1.26 | 0.0601 (0.0507–0.0642) | 1.27 |
| | 10 | 6 | 0.0594 (0.0332–0.0797) | 2.40 | 0.0585 (0.0288–0.0903) | 3.13 | 0.0220 (-0.0470–0.0709) | ND | 0.0476 (0.0252–0.0665) | 2.63 |
| | 10 | 20 | 0.0608 (0.0562–0.0652) | 1.16 | 0.0614 (0.0524–0.0656) | 1.25 | 0.0113 (0.0038–0.0414) | 10.8 | 0.0483 (0.0427–0.0525) | 1.23 |
| (e) | 4 | 6 | 0.0641 (0.0177–0.1213) | 6.87 | 0.0508 (0.0095–0.1044) | 11.0 | 0.0544 (0.0127–0.1045) | 8.23 | 0.0581 (0.0148–0.1021) | 6.90 |
| | 4 | 20 | 0.0695 (0.0469–0.0764) | 1.63 | 0.0534 (0.0375–0.0596) | 1.59 | 0.0568 (0.0408–0.0624) | 1.53 | 0.0603 (0.0432–0.0648) | 1.50 |
| | 6 | 6 | 0.0527 (0.0097–0.0920) | 9.51 | 0.0526 (0.0143–0.0849) | 5.95 | 0.0423 (0.0121–0.0712) | 5.87 | 0.0459 (0.0120–0.0783) | 6.53 |
| | 6 | 20 | 0.0562 (0.0411–0.0644) | 1.57 | 0.0527 (0.0378–0.0576) | 1.53 | 0.0444 (0.0324–0.0483) | 1.49 | 0.0454 (0.0353–0.0499) | 1.41 |
| | 10 | 6 | 0.0348 (0.0114–0.0573) | 5.05 | 0.0310 (0.0098–0.0602) | 6.16 | 0.0253 (0.0087–0.0418) | 4.81 | 0.0260 (0.0108–0.0460) | 4.24 |
| | 10 | 20 | 0.0376 (0.0298–0.0403) | 1.35 | 0.0374 (0.0239–0.0398) | 1.66 | 0.0247 (0.0204–0.0276) | 1.35 | 0.0260 (0.0208–0.0283) | 1.36 |
| (f) | 4 | 6 | 0.0667 (0.0240–0.1174) | 4.90 | 0.0520 (0.0138–0.1058) | 7.68 | 0.0573 (0.0170–0.1083) | 6.39 | 0.0587 (0.0230–0.1012) | 4.41 |
| | 4 | 20 | 0.0703 (0.0517–0.0764) | 1.48 | 0.0575 (0.0393–0.0625) | 1.59 | 0.0632 (0.0455–0.0684) | 1.50 | 0.0599 (0.0493–0.0651) | 1.32 |
| | 6 | 6 | 0.0585 (0.0237–0.0928) | 3.92 | 0.0551 (0.0160–0.0901) | 5.61 | 0.0465 (0.0189–0.0711) | 3.76 | 0.0496 (0.0197–0.0786) | 3.99 |
| | 6 | 20 | 0.0588 (0.0465–0.0659) | 1.42 | 0.0536 (0.0436–0.0605) | 1.39 | 0.0474 (0.0399–0.0519) | 1.30 | 0.0523 (0.0395–0.0562) | 1.42 |
| | 10 | 6 | 0.0364 (0.0105–0.0616) | 5.85 | 0.0331 (0.0110–0.0620) | 5.59 | 0.0265 (0.0093–0.0460) | 4.95 | 0.0320 (0.0122–0.0622) | 5.12 |
| | 10 | 20 | 0.0380 (0.0331–0.0411) | 1.24 | 0.0380 (0.0244–0.0403) | 1.65 | 0.0256 (0.0230–0.0281) | 1.22 | 0.0403 (0.0246–0.0444) | 1.81 |

*(Continued)*

**Table 4.** (Continued)

| (B) | | | | | | | | | | |
|---|---|---|---|---|---|---|---|---|---|---|
| regression algorithm | spiking levels | samples/ level | carbamazepine | | fluconazole | | lamotrigine | | levetiracetam | |
| | | | median (range) | NNI (%) | median (range) | NNI (%) | median (range) | NNI (%) | median (range) | NNI (%) |
| (a) | 4 | 6 | 0.0230 (-1.203–1.1617) | 51.8 | 0.1180 (-0.3171–0.5861) | 80.6 | 0.3336 (-0.7953–1.6524) | 79.0 | 0.0638 (-0.9038–0.7575) | 61.7 |
| | 4 | 20 | 0.0025 (-0.3535–0.4951) | 51.1 | 0.1342 (-0.0183–0.2558) | 99.8 | 0.3599 (-0.0597–0.6877) | 99.8 | 0.0343 (-0.1064–0.3030) | 76.7 |
| | 6 | 6 | 0.0252 (-0.1104–0.1743) | 68.4 | 0.0112 (-0.0482–0.0807) | 70.6 | 0.0399 (-0.1107–0.1794) | 69.1 | 0.0195 (-0.1177–0.1421) | 66.4 |
| | 6 | 20 | 0.0256 (-0.0030–0.0845) | 99.9 | 0.0129 (-0.0007–0.0426) | >99.9 | 0.0249 (0.0083–0.0872) | 100 | 0.0196 (-0.0055–0.0731) | 99.7 |
| | 10 | 6 | -0.0068 (-0.0518–0.0417) | 33.0 | -0.0381 (-0.0051–0.0281) | 32.6 | -0.0109 (-0.0409–0.0263) | 16.4 | -0.1693 (-0.3350–0.0692) | 0.491 |
| | 10 | 20 | -0.0075 (-0.0164–0.0079) | 4.12 | -0.0042 (-0.0137–0.0041) | 3.92 | -0.0103 (-0.0161–0.0001) | 0.00 | -0.1720 (-0.1966 - -0.1217) | 0.00 |
| (b) | 4 | 6 | 0.0141 (-0.6640–0.6228) | 52.6 | -0.0394 (-0.2466–0.1763) | 24.7 | -0.1046 (-0.6713–0.4679) | 29.1 | -0.0101 (-0.3010–0.4728) | 46.6 |
| | 4 | 20 | -0.0045 (-0.1500–0.2355) | 46.2 | -0.0562 (-0.1005–0.0184) | 1.09 | -0.1733 (-0.2661–0.0563) | 1.68 | 0.0110 (-0.0828–0.8615) | 68.6 |
| | 6 | 6 | 0.0036 (-0.0827–0.0946) | 55.5 | 0.0062 (-0.0299–0.0430) | 73.3 | 0.0037 (-0.0704–0.0911) | 56.0 | 0.0138 (-0.0536–0.0808) | 77.0 |
| | 6 | 20 | 0.0014 (-0.0135–0.0280) | 61.3 | 0.0084 (-0.0009–0.0154) | >99.9 | 0.0055 (-0.0101–0.0263) | 85.9 | 0.0160 (0.0017–0.0317) | 100 |
| | 10 | 6 | 0.0006 (-0.0330–0.0305) | 52.5 | 0.0079 (-0.0087–0.0234) | 94.1 | 0.0097 (-0.0098–0.0259) | 87.1 | 0.0806 (-0.0221–0.1633) | 99.9 |
| | 10 | 20 | 0.0003 (-0.0065–0.0110) | 54.9 | 0.0072 (0.0046–0.0127) | 100 | 0.0083 (0.0056–0.0142) | 100 | 0.0812 (0.0612–0.1029) | 100 |
| (c) | 4 | 6 | not evaluated | | | | | | | |
| | 4 | 20 | not evaluated | | | | | | | |
| | 6 | 6 | 0.0142 (-0.0557–0.0878) | 75.5 | 0.0085 (-0.0235–0.0331) | 79.3 | -0.0045 (-0.0797–0.0704) | 41.4 | 0.0081 (-0.0446–0.0538) | 68.3 |
| | 6 | 20 | 0.0149 (0.0008–0.0319) | 100 | 0.0085 (0.0010–0.0172) | 100 | -0.0029 (-0.0162–0.0165) | 26.8 | 0.0074 (-0.0018–0.0225) | 99.6 |
| | 10 | 6 | 0.0007 (-0.0085–0.0106) | 58.0 | 0.0001 (-0.0069–0.0078) | 51.9 | 0.0026 (-0.0052–0.0112) | 83.3 | 0.0013 (-0.0218–0.0296) | 56.1 |
| | 10 | 20 | 0.0003 (-0.0011–0.0037) | 70.4 | 0.0001 (-0.0014–0.0028) | 62.6 | 0.0026 (0.0007–0.0051) | 100 | 0.0022 (-0.0043–0.0108) | 89.1 |
| (d) | 4 | 6 | 0.0060 (0.0006–0.0109) | 100 | 0.0011 (0.0000–0.0029) | 100 | 0.0010 (0.0006–0.0110) | 100 | 0.0028 (0.0004–0.0054) | 100 |
| | 4 | 20 | 0.0063 (0.0039–0.0068) | 100 | 0.0016 (0.0008–0.0017) | 100 | 0.0010 (0.0006–0.0011) | 100 | 0.0029 (0.0020–0.0032) | 100 |
| | 6 | 6 | 0.0007 (0.0000–0.0016) | 100 | 0.0014 (0.0009–0.0015) | 100 | 0.0005 (0.0000–0.0008) | 99.8 | 0.0005 (0.0000–0.0009) | 100 |
| | 6 | 20 | 0.0008 (0.0004–0.0009) | 100 | 0.0014 (0.0010–0.0015) | 100 | 0.0005 (0.0004–0.0006) | 100 | 0.0006 (0.0004–0.0006) | 100 |
| | 10 | 6 | 0.0001 (0.0000–0.0002) | 100 | 0.0001 (0.0000–0.0001) | 100 | 0.0018 (0.0001–0.0062) | 100 | 0.0006 (0.0001–0.0022) | 100 |
| | 10 | 20 | 0.0001 (0.0000–0.0001) | 100 | 0.0001 (0.0000–0.0001) | 100 | 0.0033 (0.0012–0.0034) | 100 | 0.0012 (0.0005–0.0012) | 100 |
| (e) | 4 | 6 | 0.0073 (-0.2136–0.3109) | 88.2 | 0.0024 (-0.0701–0.0874) | 82.1 | 0.0033 (-0.1579–0.2144) | 85.5 | 0.0038 (-0.1777–0.2662) | 84.6 |
| | 4 | 20 | 0.0065 (-0.0180–0.1319) | 99.9 | 0.0060 (-0.0146–0.0220) | 98.0 | 0.0219 (-0.0107–0.0732) | 99.7 | 0.0045 (-0.0006–0.0959) | >99.9 |
| | 6 | 6 | 0.0048 (-0.0183–0.0477) | 95.0 | 0.0043 (-0.0060–0.0303) | 98.4 | 0.0036 (-0.0180–0.0555) | 92.9 | 0.0046 (-0.0153–0.0367) | 97.7 |
| | 6 | 20 | 0.0064 (0.0006–0.0223) | 100 | 0.0058 (0.0014–0.0184) | 100 | 0.0059 (0.0005–0.0097) | 100 | 0.0082 (0.0006–0.0171) | 100 |
| | 10 | 6 | 0.0012 (-0.0007–0.0044) | >99.9 | 0.0014 (-0.0006–0.0041) | 99.8 | 0.0021 (0.0003–0.0063) | 100 | 0.0040 (0.0003–0.0114) | 100 |
| | 10 | 20 | 0.0013 (0.0001–0.0019) | 100 | 0.0015 (0.0009–0.0019) | 100 | 0.0030 (0.0014–0.0039) | 100 | 0.0056 (0.0029–0.0067) | 100 |
| (f) | 4 | 6 | 0.0292 (-0.6368–0.6026) | 71.5 | 0.0017 (-0.1590–0.1824) | 57.2 | 0.0141 (-0.4067–0.4505) | 64.6 | 0.0203 (-0.1064–0.0862) | 68.6 |
| | 4 | 20 | 0.0268 (-0.1558–0.1669) | 79.7 | 0.0060 (-0.0168–0.0233) | 90.1 | 0.0223 (-0.0268–0.0711) | 99.2 | 0.0252 (-0.1003–0.1079) | 87.8 |
| | 6 | 6 | 0.0061 (-0.0615–0.0854) | 89.5 | 0.0050 (-0.0285–0.0424) | 95.4 | 0.0043 (-0.0760–0.0743) | 83.4 | 0.0066 (-0.0504–0.0764) | 93.5 |
| | 6 | 20 | 0.0124 (-0.0059–0.0415) | >99.9 | 0.0114 (0.0084–0.0226) | 100 | 0.0108 (0.0014–0.0187) | 100 | 0.0126 (0.0020–0.0372) | 100 |
| | 10 | 6 | 0.0013 (-0.0016–0.0060) | 99.8 | 0.0015 (-0.0016–0.0057) | 99.5 | 0.0021 (-0.0007–0.0065) | >99.9 | 0.0051 (-0.0033–0.0212) | >99.9 |
| | 10 | 20 | 0.0013 (0.0005–0.0022) | 100 | 0.0015 (0.0006–0.0023) | 100 | 0.0033 (0.0021–0.0036) | 100 | 0.0058 (0.0028–0.0114) | 100 |

The ranges of the calculated constant coefficients (intercepts) of the assay error equations, which correspond to the estimated SD's of the blanks, are presented in Table 4B. The proportion of *negative* intercepts observed within "small" and "large" sample sets were 19.4–99.5% and 0–100% [algorithm (a)], 0.1–75.3% and 0–98.8% [algorithm (b)], 16.7–48.6% and 0–73.2% [algorithm (c)], 0–17.9% and 0–2.0% [algorithm (e)], further, <0.1–42.8% and 0–20.3% [algorithm(f)]. Increasing the size of the sample set typically, but not always, improved the proportion of non-negative intercepts, regardless of the regression algorithm applied. Unweighted linear least squares regression [algorithm (a)] gave rise to unacceptably large proportions of negative intercepts, except for a single experimental setup (M = 6, R = 20). On the contrary, all intercepts were non-negative when $1/x^2$-weighted linear least squares regression[algorithm (d)] was performed, regardless of the number of samples and spiking levels [except for a single case with LAM (M = 6, R = 6)], or when Theil's regression [algorithms (e) and (f)] was applied to "large" sample sets, using 6 or 10 spiking levels (with a single negative intercept recorded for CBZ, M = 6).

Considering nonlinear regression, proportions of non-negative intercepts exceeding 99.9% were obtained by applying 2nd-order polynomial regression [algorithm (b)] only for FLU and LEV in the "large" sample sets, and when 6 or 10 spiking levels were used. Further cases with acceptable proportions of non-negative intercept were recorded with FLU (94.1%) and LEV (99.9%) in "small" sample sets when 10 spiking levels were employed. These proportions were unacceptably large in all other cases.

The results obtained with 3rd-order polynomial regression were inconsistent, with an unacceptably large proportion of negative intercepts recorded for all (M = 6 or 10, R = 6) or most (M = 10, R = 20) assayed drugs. In a single setup (M = 6, R = 20), 99.6–100% of the experiments with 3 of the 4 drugs gave rise to non-negative intercepts.

## Evaluation of the final precision profiles

The final precision profiles and the fitted AEE's are shown in Figs 4 and 5. The summarizing data of the AEE's and the NSSR values indicating the goodness of fit are presented in Table 5 (individual predicted SD's are provided in S4–S7 Tables). None of the linear regression algorithms yielded negative linear coefficients. Negative constant coefficients (intercepts), however, were obtained for CBZ, LAM and LEV using unweighted linear [algorithm (a)], and, for CBZ, FLU and LAM using unweighted 2nd-order polynomial least squares regression [algorithm (b), Table 5]. Based on the NSSR values, superior linear fits were obtained when the AEE's were established using Theil's regression with or without the Siegel estimator (2.492–25.66 and 2.197–24.28, respectively). $1/x^2$-weighted linear least squares and unweighted 3rd-order nonlinear regression demonstrated similar performance (NSSR = 9.021–23.11 and 4.234–10.86, respectively). Unweighted linear and 2nd-order least squares regression, on the other hand, were far less suitable, as their NSSR's ranged between 3.288–3782 and 15.88–49848, respectively.

## Discussion

The present work was conducted to provide a new methodology for the estimation of numeric SD values for each concentration level obtained in clinical LC-MS/MS drug assays. This methodology is of importance as maximally useful clinical pharmacokinetic models employed for accomplishing optimal patient care require the use of the SD reasonably associated with each assay result instead of the coefficient of variation.

As a major difference from earlier works where assay error polynomials were evaluated by running patient specimens submitted for TDM in replicates [11,15], in this study, multiple

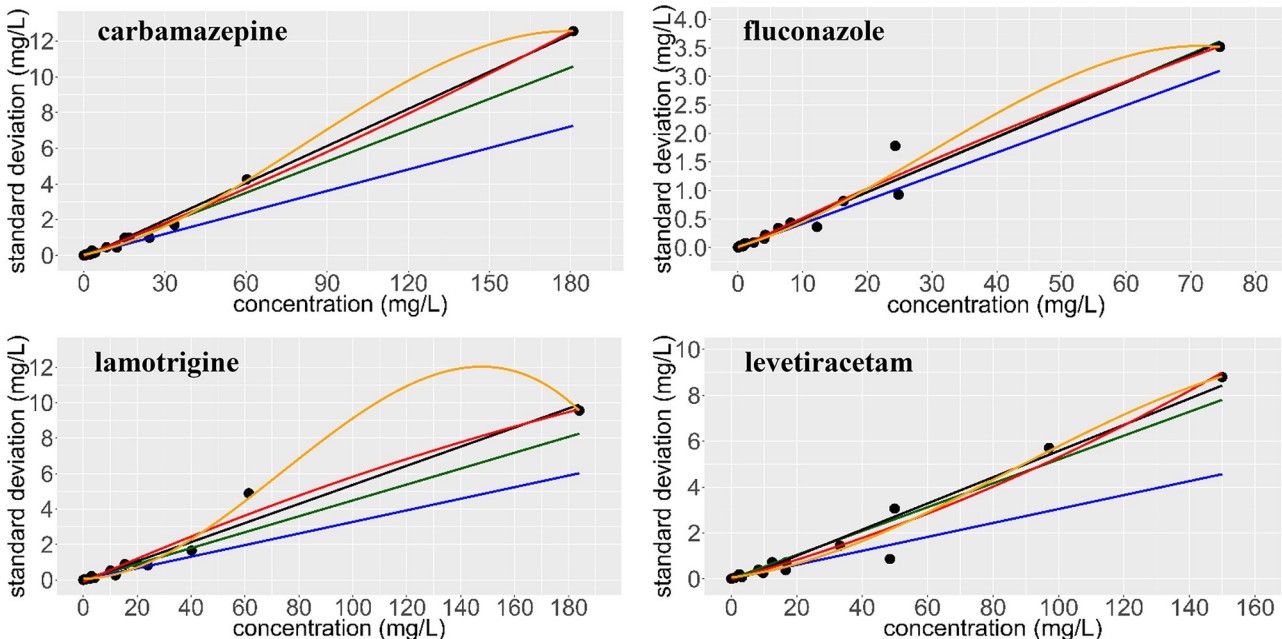

**Fig 4. Regression plots obtained by applying various regression algorithms to the precision profiles of carbamazepine, fluconazole, lamotrigine and levetiracetam and by using all of the serum specimens (N = 24) and spiking levels (M = 20+blank) of the 3 experiments.** Each data point displays the standard deviation of the assay results. Green line: Theil's regression, black line: unweighted linear least squares, blue line: $1/x^2$-weighted linear least squares, red curve: unweighted 2$^{nd}$-order least squares, orange curve: unweighted 3$^{rd}$-order least squares regression. See Table 5 for a detailed evaluation.

independent blank human serum specimens were spiked with the analytes at known levels and evaluated in a single run. Because patient samples are typically assayed in a single measurement in routine diagnostics, along with the technical difficulties of presenting samples to cover the entire assayed concentration range, and with the lack of knowledge of nominal analyte concentrations in the real samples, we propose that the presented approach is more feasible and correct for the development of AEE's. An additional benefit is that this approach can be applied in parallel with experiments performed to establish method accuracy and precision, allowing AEE's to become a tool of method performance evaluation and part of the validation process.

Measures were taken to simulate special clinical situations. In experiments 1 and 2, 75% of the serum specimens employed had a positive lipemia-icterus-hemolysis (LIH) index, a condition frequently encountered and likely to interfere with assay performance. In experiment 1, a proportion of the spiking levels would be considered as supratherapeutic according to the widely considered therapeutic concentration ranges. In experiment 2 and 3, some of the spiking levels were lower than the lowest level calibrator and would therefore not be reported by most clinical laboratories.

The number of samples in the "small" sample sets was selected to make it equal to that recommended as a minimum by the bioanalytical method validation guideline of the European Medicines Agency [18]. The number of samples in the "large" sample sets was chosen to allow a low fractional error of the estimate of the SD while still providing us with a sufficiently large number of combinations of samples. This was based on the work of Ahn and Fessler who described the relationship between the error of the estimated assay SD and the number of specimens as $\frac{\sigma_R S}{\sigma} \approx \frac{1}{\sqrt{2(R-1)}}$, where $\sigma_R S/\sigma$ is the fractional error of the estimate of $\sigma$, the true value of the standard deviation [24]. The error of the SD is 32% when R = 6 and 16% when R = 20.

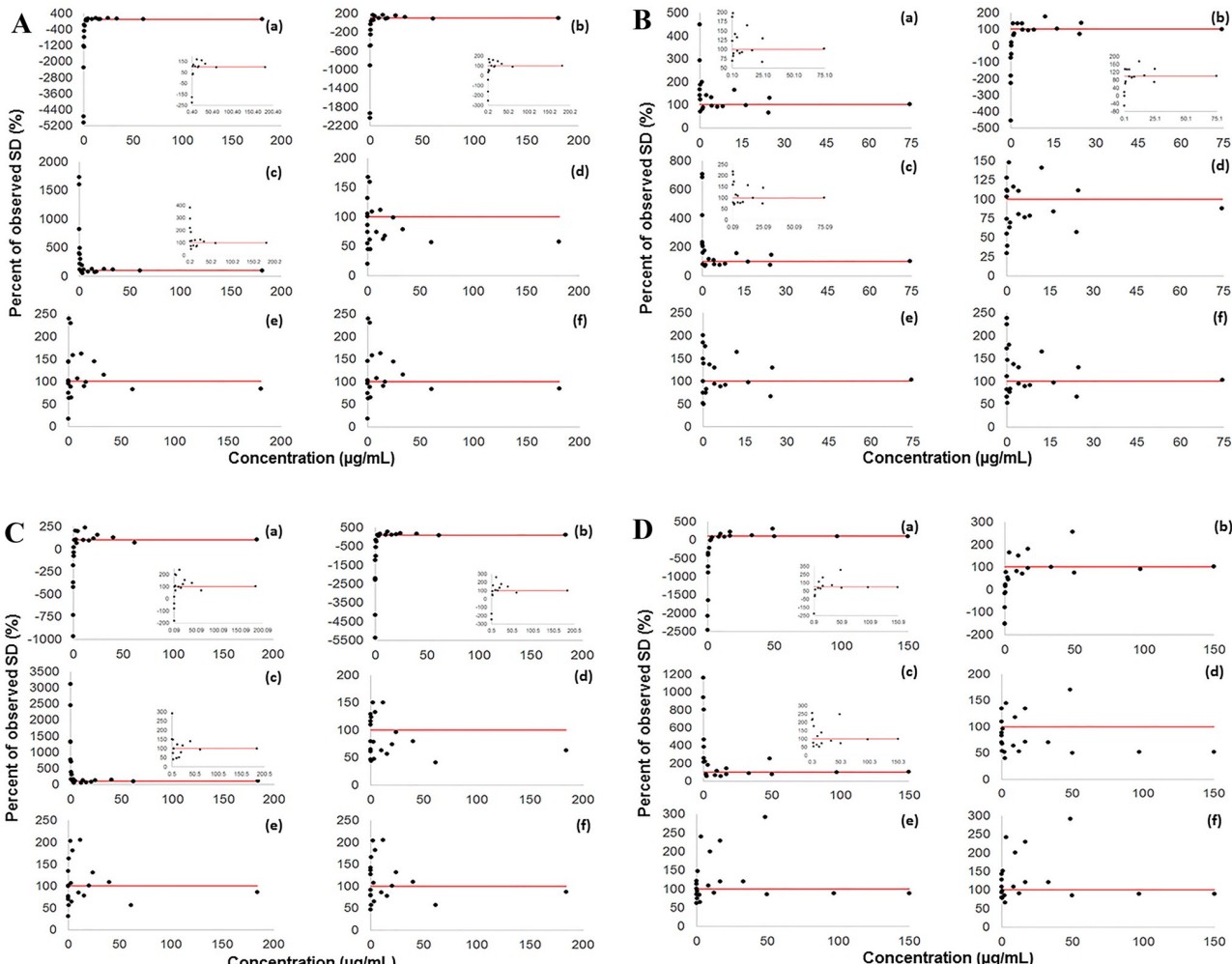

**Fig 5. Accuracy of the prediction of SD's by applying various regression algorithms to the final precision profiles of (A) carbamazepine, (B) fluconazole, (C) lamotrigine and (D) levetiracetam (M = 20 spiking levels+blank, N = 24 specimens/spiking level).** Red lines indicate perfect (100%) agreement between predicted and experimentally determined SD's. Insets show results for concentration ranges in which the inaccuracy of the predictions was on the same scale. The applied regression methods are (a) unweighted linear least squares, (b) unweighted $2^{nd}$-order least squares, (c) unweighted $3^{rd}$-order least squares, (d) $1/x^2$-weighted linear least squares, (e) Theil's regression, (f) Theil's regression with the Siegel estimator.

The evaluation of the performance of the AEE's obtained using various regression algorithms needs to be based on specific considerations. First, because SD's can only take positive values, and a key goal of developing AEE's is to have a tool for the estimation of assay precision all the way down to zero concentration, the primary aspect of judging the applicability of a regression algorithm is whether negative constant coefficients (intercepts) occur or not. Our results demonstrate that, regardless of the order of the fitted polynomial, the size of the sample set or the number of spiking levels, unweighted least squares regression yields negative intercepts in many sample sets and is, therefore, not suitable when LC-MS/MS is employed. Second, the characterization of the goodness of fit employing correlation coefficients is not helpful for this task as the relationship between concentrations and experimentally determined SD's is unlikely to be very strong, correlation coefficients are subject to the presence of even a single outlier SD, and provide no information on the potential bias of the regression. We therefore propose the use of the sum of squared residuals normalized to the predicted value (NSSR),

**Table 5. Regression equations and the normalized sums of squared residuals obtained applying various regression algorithms to the combined results of the 3 experiments performed on carbamazepine, fluconazole, lamotrigine and levetiracetam (N = 24 specimens/spiking level, M = 20 spiking levels+blank).**

| | carbamazepine | fluconazole | lamotrigine | levetiracetam |
|---|---|---|---|---|
| | | regression equations | | |
| unweighted linear least squares | y = - 0.1019 + 0.0692x | y = 0.007963 + 0.04816x | y = - 0.02029 + 0.05383x | y = - 0.1389 + 0.05711x |
| unweighted $2^{nd}$ degree least squares polynomial | y = - 0.04192 + 0.05999x + 0.00005414$x^2$ | y = - 0.0001215 + 0.05402x - 0.0000872$x^2$ | y = - 0.01097 + 0.06685x - 0.00007571$x^2$ | y = 0.0102636 + 0.03936x + 0.0001376$x^2$ |
| unweighted $3^{rd}$ degree least squares polynomial | y = 0.03590 + 0.03182x + 0.0008160x2–0.000003368$x^3$ | y = 0.01932 + 0.02799x + 0.001470x2–0.00001631$x^3$ | y = 0.06309 + 0.003122x + 0.001601$x^2$–0.000007272$x^3$ | y = 0.06295 + 0.01799x + 0.0006362$x^2$–0.000002445$x^3$ |
| $1/x^2$-weighted linear least squares | y = 0.001645 + 0.040093x | y = 0.001567 + 0.04267x | y = 0.002221 + 0.032688x | y = 0.005912 + 0.030379x |
| Theil's linear | y = 0.001423 + 0.058411x | y = 0.002558 + 0.04894x | y = 0.001511 + 0.04483x | y = 0.004113 + 0.05199x |
| Theil's linear with Siegel estimator | y = 0.001459 + 0.056877x | y = 0.003714 + 0.05145x | y = 0.002215 + 0.04562x | y = 0.005305 + 0.054723x |
| | | normalized sums of squared residuals | | |
| unweighted linear least squares | 20.87 | 3.288 | 55.59 | 3782 |
| unweighted $2^{nd}$ degree least squares polynomial | 28.92 | 49848 | 15.88 | 186.39 |
| unweighted $3^{rd}$ degree least squares polynomial | 6.829 | 4.234 | 10.86 | 7.400 |
| $1/x^2$-weighted linear least squares | 23.11 | 11.00 | 9.118 | 9.021 |
| Theil's linear | 25.66 | 3.648 | 7.829 | 2.492 |
| Theil's linear with Siegel estimator | 24.28 | 3.051 | 4.271 | 2.197 |

an unbiased indicator for comparing the goodness-of-fit of the AEE's obtained using various types of regression or in various experiments.

The NSSR value is not the only indicator of the performance of the applied regression algorithms. The key advantage of applying algorithms (d)-(f) becomes apparent by comparing the estimates of SD obtained at low and at high concentration levels. The percentage errors of the estimates of SD obtained at low levels using unweighted least squares regression [algorithms (a)-(c)] were unacceptably large (Fig 5). In contrast, algorithms (d)-(f) yielded SD estimates on the same scale throughout the entire concentration range, down to zero concentration. Since making acceptable estimates at low concentrations (i.e. those outside the validated concentration range) and at the blank are of exceptional importance for avoiding the loss of pharmacokinetic information, the consideration of algorithms (d)-(f) is recommended (S4–S7 Tables).

The selection of the various regression equations was based on earlier work of Jelliffe and Tahani who demonstrated that precision profiles obtained on clinical analyzers could often be characterized by unweighted second degree polynomials [11]. As shown above, the LC-MS/MS method described herein was characterized by linear precision profiles. Based on theoretical considerations and empirical findings, Gu et al proposed that SD is directly proportional to the analyte concentration in all bioanalytical assays performed using LC-MS/MS, and 1/concentration$^2$ is the correct weight for calibration when LC-MS/MS is used [8]. Although the reasoning Gu et al provide is short of comprehensive theoretical evidence, $1/x^2$ weighting of quantitative relationships has been proposed in several publications on bioanalytical methodology, rendering reasonable the hypothesis that it can also be used for establishing precision profiles. Our results confirm that $1/x^2$-weighted linear least squares regression consistently provides better estimates of the SD than unweighted linear or nonlinear least squares

regression algorithms do. The disadvantage of this approach, in addition to the lack of comprehensive theoretical evidence to substantiate its use for developing AEE's, is its sensitivity to outliers in the precision profile which can be overcome by assessing an adequate number of samples and spiking levels.

Theil's regression [algorithm (e)] is a very robust, nonparametric median-based, unbiased linear regression approach without any statistical assumptions and easy to calculate using computer software. When a sufficient amount of data pairs is available, Theil's regression gives rise to the true quantitative relationship between the drug concentration and the SD as long as this relationship is linear. Its application afforded acceptable estimates of SD's throughout the assayed concentration ranges, was theoretically sound, and allowed the tolerance of the outliers encountered (which does not mean that it makes good estimates for outliers). While Theil's regression method is relatively insensitive to the presence of outlier SD's in the precision profile, it eventually becomes biased when $\geq$29.4% of the data points are outliers. This limitation is overcome by the use of the Siegel estimator which is maximally robust [19,20,22]. Earlier, the utility of Theil's regression has been demonstrated for a variety of bioanalytical tasks [25,26]. In our experiments, Theil's regression was found to yield superior estimates of assay precision all the way down to zero concentration (Fig 5), with positive intercepts for almost all combinations of specimens when precision profiles established using "large" sample sets were evaluated (Table 4B). We therefore recommend using Theil's regression with the Siegel estimator as the first choice for establishing AEE's of TDM assays when the relationship between drug concentrations and SD's is linear, while $1/x^2$-weighted linear least-squares can be an efficient, though not universal, alternative.

Our results confirm experimentally that the precise estimation of the coefficients of the AEE's requires the use of large serum sample sets, while the differences observed in the ranges of the linear coefficients obtained with "small" and "large" sample sets in this study can be well explained by the findings of Ahn and Fessler [24]. Our results also demonstrate that the quantitative relationship depends on the number of the spiking levels used. The evaluation of accuracy and precision results obtained in 5 or 6 independent samples at 4 spiking levels, as proposed by the widely employed bioanalytical method validation guidelines, is clearly suboptimal in this respect [18,27].

In experiments 1 and 2, where a substantial proportion of the sample sets consisted of hemolytic, hyperbilirubinemic and lipemic serum specimens, the median linear coefficients obtained were consistently higher than those obtained in experiment 3 where"normal" serum was used. While these results do not imply that serum specimens containing these interferences should be rejected, the impact of the above interferences on the analysis is apparent, even when prepared samples are diluted substantially before analysis. It is therefore crucial to consider the inclusion of an adequate number of specimens with these interferences in serum sample sets employed for developing AEE's to avoid obtaining falsely low estimates of SD's.

The custom of not reporting the correct mathematical credibility of each measurement causes clinical and bioanalytical laboratories to censor many clinically important "below-LLOQ" results. The general recommendation that establishing concentration-dependent estimates of SD allows the laboratory to correctly report analytical results lower than the LLOQ for PK modeling, made by Jelliffe et al earlier, is confirmed by our experiments. The reported concentration range can be extended all the way down to zero concentration with acceptable accuracy and precision for pharmacokinetic modeling when the employed bioanalytical method relies on the use of LC-MS/MS [15].

The approach presented here can be incorporated flawlessly into the workflow of bioanalytical method validation according to the guidelines of the European Medicines Agency or the US Food and Drug Administration [18,27]. In terms of controlling precision, current

guidelines focus only on the prevention of the coefficient of variation (CV%) becoming "unacceptably" great at the low end of the assay range by applying the arbitrary concept of LLOQ. Employing such arbitrary limits of "acceptability" during the evaluation of method accuracy and precision is a debatable concept in general, and is clearly not useful for incorporating analytical results correctly in pharmacokinetic models and practical patient care [28–30]. Moreover, current tools proposed by the guidelines for monitoring the performance of bioanalytical methods, such as running internal quality controls, incurred sample reanalysis, and participation in external quality assessment schemes, focus only on accuracy but not on precision.

One may argue that the development of an assay error equation leads to the ignorance of the between-run performance of the analytical method. Between-run experimens are usually conducted by assaying the same specimen spiked to various levels in 3–6 batches, often on separate days. These specimens are usually not retained after method validation has been completed, instead, during routine clinical use, it is the internal controls measured in each batch which provide similar information, monitored using process control charts. Therefore it has been proposed that, in view of the diversity of random events contributing to the assay error, the information provided by between-run experiments should be incorporated into the precision profile established for developing the assay error equation [15]. The methodology we describe in the present work does not imply that validation assays should not be peformed on separate days. Our results in fact demonstrate that performing multiple independent experiments using different serum specimens each time increases the credibility of the AEE's. Nevertheless, the integrated evaluation of the validation measurements renders the separation of within-run and between-run validation tests unnecessary, while providing quantitative data on precision across the assayed concentration range, considerably exceeding the quality of the information delivered by within-run and between-run validation tests performed as recommended by [18] and [27].

The experimental setup described here can be extended to studying the impact of further pre-analytical factors, such as storage conditions. At the same time, the laboratory is free to incorporate further data into the developed AEE's, as well as to develop new ones for comparisons, whenever it is desirable to check the assay for drift or other departure from previous behavior.

## Conclusions

Software-guided adaptive control of drug therapy for the maximally precise achievement of clinically selected targets can now be carried out for each individual patient. Consequently, integrating correct knowledge about the performance of the analytical method offers important opportunities for improvement for clinical laboratories. An essential element of this process is the correct estimation of analytical precision and weighting of each measured concentration to obtain proper drug dosage adjustments.

The proposed paradigm of evaluating clinical and bioanalytical drug assays for this purpose consists of the following.

1. Reporting the measurement along with its estimated SD so the result can be incorporated into population pharmacokinetic models to obtain correct parameter estimates without which maximally precise initial dosage regimens cannot be developed, and into the subsequent individual models used for maximally precise dosage adjustment for each individual patient.

2. Establishing assay error equations specifically for the method employed in multiple experiments by using several spiking levels and by spiking a large number of (preferably, at least

20) independent serum specimens at each level. When the precision profile is linear, as in most LC-MS/MS assays, it can be estimated efficiently using Theil's regression with the Siegel estimator all the way down to zero concentration, eliminating the LLOQ and any need to censor low results. Full knowledge about the results obtained over the entire assay range is now available to laboratory personnel, to clinicians and to pharmacotherapists.

## Supporting information

**S1 Table. Preparation and analyte content of the employed calibrator samples.** CBZ, carbamazepine. FLU, fluconazole. LAM, lamotrigine. LEV, levetiracetam. SS1, Stock solution 1.
(DOCX)

**S2 Table. Preparation and analyte content of spiked independent serum samples.**
CBZ-SS1, carbamazepine stock solution 1. FLU-SS1, fluconazole stock solution 1. LAM-SS1, lamotrigine stock solution 1. LEV-SS1, levetiracetam stock solution 1. CBZ-SS2, carbamazepine stock solution 1. FLU-SS2, fluconazole stock solution 1. LAM-SS2, lamotrigine stock solution 1. LEV-SS2, levetiracetam stock solution 1.
(DOCX)

**S3 Table. Matrix factors and internal standard-corrected matrix factors.**
(DOCX)

**S4 Table. Observed versus predicted standard deviations of carbamazepine.** OLS, unweighted linear least squares. WLS, $1/x^2$-weighted linear least squares.
(DOCX)

**S5 Table. Observed versus predicted standard deviations of fluconazole.** OLS, unweighted linear least squares. WLS, $1/x^2$-weighted linear least squares.
(DOCX)

**S6 Table. Observed versus predicted standard deviations of lamotrigine.** OLS, unweighted linear least squares. WLS, $1/x^2$-weighted linear least squares.
(DOCX)

**S7 Table. Observed versus predicted standard deviations of levetiracetam.** OLS, unweighted linear least squares. WLS, $1/x^2$-weighted linear least squares.
(DOCX)

**S1 File. Zipped data files employed for performing the combinatorial calculations.**
(ZIP)

**S2 File. Zipped data files employed for creating the final precision profiles.**
(ZIP)

**S3 File. Scripts, written in R, employed for performing the calculations.**
(DOCX)

## Acknowledgments

The authors gratefully acknowledge the careful and thoughtful help of Dr. David Bayard in the preparation of this manuscript. Tünde Heiger-Holczer is acknowledged for managing serum specimens. Katalin Rischák and Róbert Farkas are acknowledged for their valuable advice on the manuscript.

## Author Contributions

**Conceptualization:** Gellért Balázs Karvaly, Michael N. Neely, Roger W. Jelliffe.

**Data curation:** Gellért Balázs Karvaly, Krisztián Kovács, István Vincze.

**Funding acquisition:** Barna Vásárhelyi.

**Investigation:** Gellért Balázs Karvaly.

**Methodology:** Gellért Balázs Karvaly.

**Project administration:** Gellért Balázs Karvaly, Krisztián Kovács, István Vincze.

**Software:** Gellért Balázs Karvaly.

**Supervision:** Michael N. Neely, Barna Vásárhelyi, Roger W. Jelliffe.

**Validation:** István Vincze.

**Visualization:** Gellért Balázs Karvaly, Krisztián Kovács.

**Writing – original draft:** Gellért Balázs Karvaly, Michael N. Neely, Barna Vásárhelyi, Roger W. Jelliffe.

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
