## [Decision Letter · Decision Letter 0]

24 Dec 2019

PONE-D-19-27122

Development of a methodology to make individual estimates of the precision of liquid chromatography-tandem mass spectrometry drug assay results for use in population pharmacokinetic modeling and the optimization of dosage regimens

PLOS ONE

Dear Dr Karvaly,

Thank you for submitting your manuscript to PLOS ONE. After careful consideration, we feel that it has merit but does not fully meet PLOS ONE’s publication criteria as it currently stands. Therefore, we invite you to submit a revised version of the manuscript that addresses the points raised during the review process.

We would appreciate receiving your revised manuscript by Feb 07 2020 11:59PM. To enhance the reproducibility of your results, we recommend that if applicable you deposit your laboratory protocols in protocols.io, where a protocol can be assigned its own identifier (DOI) such that it can be cited independently in the future. For instructions see: http://journals.plos.org/plosone/s/submission-guidelines#loc-laboratory-protocols

We look forward to receiving your revised manuscript.

Kind regards,

Jed N. Lampe, Ph.D.

Academic Editor

PLOS ONE

Journal Requirements:

Reviewers' comments:

Reviewer's Responses to Questions

**Comments to the Author**

1. Is the manuscript technically sound, and do the data support the conclusions?

Reviewer #1: Yes

2. Has the statistical analysis been performed appropriately and rigorously? 

Reviewer #1: Yes

3. Have the authors made all data underlying the findings in their manuscript fully available?

Reviewer #1: Yes

4. Is the manuscript presented in an intelligible fashion and written in standard English?

Reviewer #1: Yes

5. Review Comments to the Author

Reviewer #1: Review of PONE-D19-27122 entitled “Development of a methodology to make individual estimates of the precision of liquid chromatography-tandem mass spectrometry drug assay results for use in population pharmacokinetic modeling and the optimization of dosage regimens” by G.B. Karvaly, et al.

Summary: In this study, the authors investigate the use of assay error equations (AEE) to characterize the relationship between concentration and standard deviation (SD) for analytical LC-MS/MS drug quantification assays. According to the authors, optimization of the assay error estimation by determination of the SD would significantly improve clinical pharmacokinetic models. The authors evaluated using a combinatorial approach several regression techniques to develop AEE’s for 3 independent measurements at 4, 6 and 10 concentration levels, respectively, and assess the effect of sample size and of lipemia-icterus-hemolysis (LIH) interferences on the method precision for 4 different drugs. The conclusions from this work were that for a LC-MS/MS linear profile the assay error can be properly estimated by using Theil’s regression with the Siegel estimator, that large sample set of 20 provides better precision estimation and that LIH plasma samples can have some impact on the LC-MS/MS analysis and should be included in the bioanalytical method evaluation.

Review: Overall, this work provides an interesting methodology to estimate assay error and thus, a deeper assessment of the precision of an analytical LC-MS/MS drug assay. However, clarity of the manuscript is lacking and making it difficult to understand some of the methodology used and resulting data.

Major concerns:

1. Numerous inconsistencies in the Materials and methods section are present and need to be reviewed:

a. In the Chemicals and solutions section, preparation of the IS working solution (page 6, line 113-117) contains more internal standard compounds (9 µg/mL; 0.9 µg/mL for CBZ) than the stock solutions (2 µg/mL; 0.2 µg/mL for CBZ)? Please review and correct.

b. The last sentence in the Serum specimens section (Page 8, line 142 “Spiking of analytes…”) belongs to the Sample preparation sections.

c. Table 1 belongs to the results section, since it is showing the within-run accuracy and precision data. Columns with the “nominal analyte concentration” tested and the “measured mean analyte concentration” should both be reported.

d. Unit used should be checked throughout the manuscript. For example, “analyte concentration” unit in Table 1 is in µg/mL, while Table S2 is showing ng/mL.

e. Details about the concentration range used for spiking and dilution performed should be provided with exact volume of spiked solutions added to the serum specimens.

f. “The calculations were performed using computer scripts described in section 2.6”. Page 9, line 174-175. Where is the section 2.6 mentioned?

g. In Table 2 (Page 11), please provide retention time obtained for each analyte evaluated.

h. No detail is provided in the manuscript about the standard curve used for this work, except for Figure 3 showing representative calibration plots for each analyte. Details about the standard curve preparation and range used should be provided in the Materials and methods section, same for the single point calibration used for lower concentrations.

i. Further details in the preparation of the sample carry-over and matrix factors should be provided. What is the concentration of the analyte in the sample carry-over? Only a weight is reported, but no volume. Preparation of the matrix factor samples is unclear and needs to be revised, particularly on how the analytes were added to the protein precipitated sample. Acetonitrile is mentioned as the solvent used for the protein precipitation, although this solvent was not mentioned in the Sample preparation section (Page 10, line 184). Please provide details accordingly.

j. Matrix factor was done using 6 blank serum specimens, but the effect of hyperbilirubinemic, hemolytic and lipemic serum specimens was not assessed. Since these specimens were used in experiments 1 and 2, they should be evaluated for matrix effect too.

2. The data from 3-independent experiments 1, 2 and 3 are combined to assess precision of the assay, however no between-run assays (or inter-day assays) was performed. The inter-day variation evaluation is mentioned in the Introduction section by the authors (Page 5, line 88-89), but was not performed in this work. This seems to be an important assessment of the analytical assay when combining independent measurements and should be addressed by the authors.

3. Figure 3 is showing the wrong chromatograms for the lamotrigine analyte (see mass transition on the left corner of the graph), please correct analyte and internal standard chromatograms.

4. As mentioned in page 12, line 226 method validation referred to the “performance of the calibration curves”, although in the Results section under the Method validation paragraph, data in Table 1 is referred. The data in Table 1 corresponds to the 3-independent experiments evaluated, it is unclear if these data corresponds to the calibration curves or not, particularly when the representative calibration curves shown in Figure 3 don’t match the concentration evaluated. This should be clarified.

Minor concerns:

-Page 4, line 64 (Introduction): “Mounting evidence suggests” should be changed to “Increasing evidences suggest”.

-Page 7, line 130 (Materials and Methods): “5366 g” should be changed to “5,366 x g”.

-Page 8, Table 1 (Materials and Methods): “number of experiment” should be changed to “experiment number”.

-Page 10, line 194 (Materials and Methods): “Analysis” should be changed to “Analytical method”.

-Number reported for the concentrations are inconsistent (e.g. 0.203 vs 0.012). Use of significant number of 3, for example, can be applied throughout the manuscript and provide consistent reporting.

-Page 14, line 262, â and ĉ are in italic in the text but not in the equations (3) and (4). Please revise and be consistent.

-Page 14, line 267, for consistency with Page 13, line 251, “performed in [Equation (4)].” should be changed to “performed in equation (4).”.

-Page 22, line 382, “efficient approach to” should be changed by “new methodology for”.

-Page 22, line 383, the whole sentence “This is of…” should be revised for its English.

-Page 22, line 385, “accompishing” should be corrected with “accomplishing”.

-Page 28, line 531, “wiithout” should be corrected.

-Page 28, line 538, please revise “full transparency” which is not the most appropriate wording here.

6. PLOS authors have the option to publish the peer review history of their article (what does this mean?). If published, this will include your full peer review and any attached files.

Reviewer #1: No

---

## [Author Response · Author response to Decision Letter 0]

3 Feb 2020

Please note that reference made to the pagination is based on the manuscript saved without tracking changes (Karvaly manuscript.docx).

Major concerns:

1. Numerous inconsistencies in the Materials and methods section are present and need to be reviewed:

a. In the Chemicals and solutions section, preparation of the IS working solution (page 6, line 113-117) contains more internal standard compounds (9 µg/mL; 0.9 µg/mL forCBZ) than the stock solutions (2 µg/mL; 0.2 µg/mL for CBZ)? Please review and correct.

The respective clarification has been made (lines 122-126).

b. The last sentence in the Serum specimens section (Page 8, line 142 “Spiking of analytes…”) belongs to the Sample preparation sections.

The respective correction has been made (lines 203-205).

c. Table 1 belongs to the results section, since it is showing the within-run accuracy and precision data. Columns with the “nominal analyte concentration” tested and the “measured mean analyte concentration” should both be reported.

The requested corrections and amendments have been made. Table 1 has been renumbered as Table 3 and is found starting with line 339. Mean measured analyte concentrations are reported. The respective column headings have been updated. 

d. Unit used should be checked throughout the manuscript. For example, “analyte concentration” unit in Table 1 is in µg/mL, while Table S2 is showing ng/mL.

The corrections of the units have been made in the respective supplementary Tables now numbered as S4, S5, S6 and S7.

e. Details about the concentration range used for spiking and dilution performed should be provided with exact volume of spiked solutions added to the serum specimens.

The requested information is provided in S1 (calibrators) and S2 (spiked independent specimens) Tables of the revised manuscript. Reference to these Supplementary tables are made in lines 199-200.

f. “The calculations were performed using computer scripts described in section 2.6”. Page 9,line 174-175. Where is the section 2.6 mentioned?

The manuscript does not have a section 2.6. Computer scripts are provided in S3 file. The respective correction has been made in the manuscript (line 190).

g. In Table 2 (Page 11), please provide retention time obtained for each analyte evaluated.

Table 2 has been renumbered as Table 1. The respective amendment has been made (starting with line 235).

h. No detail is provided in the manuscript about the standard curve used for this work, except for Figure 3 showing representative calibration plots for each analyte. Details about the standard curve preparation and range used should be provided in the Materials and methods section, same for the single point calibration used for lower concentrations.

S1 Table of the revised manuscript provides the requested information on the preparation of and the concentration ranges covered by the calibrators employed. S1 Table has been added, with S1 Table of the original manuscript renumbered as S3 Table. Clarification on single-point calibration has also been made (lines 220-221).

i. Further details in the preparation of the sample carry-over and matrix factors should be provided. What is the concentration of the analyte in the sample carry-over? Only a weight is reported, but no volume. Preparation of the matrix factor samples is unclear and needs to be revised, particularly on how the analytes were added to the protein precipitated sample. Acetonitrile is mentioned as the solvent used for the protein precipitation, although this solvent was notmentioned in the Sample preparation section (Page 10, line 184). Please provide details accordingly.

The respective corrections have been made. Details on the preparation of the sample carry-over solution are provided in Materials and methods, Chemicals and solutions, lines 127-129. Details on the preparation of solutions employed in the matrix factor study are given in the same section, lines 130-135. The concentration of analytes in the carry-over solution is displayed in Materials and methods, Method validation, lines 253-256. Preparation of the matrix factor samples is described in the same section, lines 257-267. The description of the preparation of the IS has been amended to include the name of the solvent (acetonitrile) in line 126.

The revision of the paper revealed that the original manuscript contained certain typos and minor mistakes. E.g. the weight of injected analytes was given as 40 pg instead of 400 pg. In the carry-over experiments internal standard carry-over was not tested. In addition, the preparation of the samples for the matrix factor study was revised and updated for performing the experiments requested by the Reviewer (see point j.).

j. Matrix factor was done using 6 blank serum specimens, but the effect of hyperbilirubinemic, hemolytic and lipemic serum specimens was not assessed. Since these specimens were used in experiments 1 and 2, they should be evaluated for matrix effect too.

The requested experiments were performed. S3 Table (renumbered from S1) contains the updated results. The conclusions of the experiments are described in lines 345-348.

2. The data from 3-independent experiments 1, 2 and 3 are combined to assess precision of the assay, however no between-run assays (or inter-day assays) was performed. The inter-day variation evaluation is mentioned in the Introduction section by the authors (Page 5, line 88-89), but was not performed in this work. This seems to be an important assessment of the analytical assay when combining independent measurements and should be addressed by the authors.

We propose that within-run and between-run assays should be integrated into establishing precision profiles. We have added a paragraph detailing this paradigm in Discussion, lines 576-592.

3. Figure 3 is showing the wrong chromatograms for the lamotrigine analyte (see mass transition on the left corner of the graph), please correctanalyte and internal standard chromatograms.

The respective corrections have been made and a new version of Figure 3 (Figure 3-revised.tif) has been uploaded.

4. As mentioned in page 12, line 226 method validation referred to the “performance of the calibration curves”, although in the Results section under the Method validation paragraph, data in Table 1 is referred. The data in Table 1 corresponds to the 3-independent experiments evaluated, it is unclear if these data corresponds to the calibration curves or not, particularly when the representative calibration curves shown in Figure 3 don’t match the concentration evaluated. This should be clarified.

The requested clarifications have been made. Table 2 in the revised manuscript contains detailed information on the performance of the calibration curves (starting with line 333). The results of the accuracy and precision studies obtained in spiked independent matrices are described more clearly in a separate paragraph (lines 335-337).

All of the accuracy and precision results have been revised. Correspondingly, minor corrections have been made to the presented numbers.

Minor concerns:

-Page 4, line 64 (Introduction): “Mounting evidence suggests” should be changed to “Increasing evidences suggest”.

The respective correction has been made (line 71).

-Page 7, line 130 (Materials and Methods): “5366 g” should be changed to “5,366 x g”.

The respective correction has been made (line 149).

-Page 8, Table 1 (Materials and Methods): “number of experiment” should be changed to “experimentnumber”.

The respective correction has been made (see Table 3).

-Page 10, line 194 (Materials and Methods): “Analysis” should be changed to “Analytical method”.

The respective correction has been made (line 216).

-Number reported for the concentrations are inconsistent (e.g. 0.203 vs 0.012). Use of significant number of 3, for example, can be applied throughout the manuscript and provide consistent reporting.

The numerical data have been modified as requested by the Reviewer. We have also revised the numerical data included in Table 3 and have corrected minor inaccuracies.

-Page 14, line 262, â and ĉ are in italic in the text but not in the equations (3) and (4). Please revise and be consistent.

The respective correction has been made (lines 288 and 289).

-Page 14, line 267, for consistency with Page 13, line 251, “performed in [Equation (4)].” should be changed to “performed in equation (4).” .

The respective correction has been made (line 295).

-Page 22, line 382, “efficient approach to” should be changed by “new methodology for”.

The respective correction has been made (line 450).

-Page 22, line 383, the whole sentence “This is of…” should be revised for its English.

The respective correction has been made (lines 451-454).

-Page 22, line 385, “accompishing” should be corrected with“accomplishing”.

The respective correction has been made (line 453).

-Page 28, line 531, “wiithout” should be corrected.

The respective correction has been made (line 611).

-Page 28, line 538, please revise “full transparency” which is not the most appropriate wording here.

The respective correction has been made (lines 619).

---

## [Editor Report · Decision Letter 1]

18 Feb 2020

Development of a methodology to make individual estimates of the precision of liquid chromatography-tandem mass spectrometry drug assay results for use in population pharmacokinetic modeling and the optimization of dosage regimens

PONE-D-19-27122R1

Dear Dr. Karvaly,

We are pleased to inform you that your manuscript has been judged scientifically suitable for publication and will be formally accepted for publication once it complies with all outstanding technical requirements.

With kind regards,

Jed N. Lampe, Ph.D.

Academic Editor

PLOS ONE

Additional Editor Comments (optional):

Please follow all PLOS One standards and guidelines when preparing your final manuscript for acceptance.
---

## [Editor Report · Acceptance letter]

20 Feb 2020

PONE-D-19-27122R1 

Development of a methodology to make individual estimates of the precision of liquid chromatography-tandem mass spectrometry drug assay results for use in population pharmacokinetic modeling and the optimization of dosage regimens 

Dear Dr. Karvaly:

I am pleased to inform you that your manuscript has been deemed suitable for publication in PLOS ONE. Congratulations! Your manuscript is now with our production department. 

With kind regards,

on behalf of

Dr. Jed N. Lampe 

Academic Editor

PLOS ONE